# A Practical Auditing Framework for World Models: Geometry, Equivariance, and Identifiability

## Abstract

Representation learning for model-based RL offers sample efficiency but raises a critical auditing question: which properties of a learned representation actually govern downstream performance, and how can we verify them without expensive retraining? We propose a practical auditing framework based on a sufficient stability bound that decomposes the suboptimality gap into three verifiable channels: geometric distortion $\kappa$, identifiability (proxied by Total Correlation), and symmetry violation (proxied by Local Equivariance Error). Crucially, this bound serves as a safety condition rather than a linear predictor, explicitly anchoring error scaling to MDP Lipschitz constants. To interpret these components, we provide two mechanistic perspectives: a quotient-space Johnson–Lindenstrauss argument explains how equivariance reduces effective dimensionality, and a geometry–equivariance trade-off quantifies why non-isometric actions inevitably increase distortion. Building on this theory, we propose a lightweight diagnostic protocol that audits existing checkpoints. Using a single calibrated constant $\beta$, our framework consistently covers the performance gap across training trajectories, offering a principled *auditing* tool distinct from architectural *design*. On DreamerV3 world models, these diagnostics are reproducible, require no retraining, and demonstrate that structural stability bounds can effectively flag failure modes even when simple correlation metrics fail.

## 1 Introduction

Learning actionable world representations is a central lever for sample-efficient RL. Yet despite strong empirical progress, it remains hard to explain *why* a given representation helps or hurts control. Practitioners often rely on indirect proxies like reconstruction error, but these metrics lack a causal link to the downstream value estimation error.

Classical analyses attempt to bridge this gap but often sit at two extremes: strict behavioral equivalence (Bisimulation) is too restrictive, while one-step prediction objectives (DeepMDP) lack global stability guarantees (Ferns et al., 2011; Gelada et al., 2019). Meanwhile, Lipschitz arguments provide clean scaling laws but offer few practitioner-ready diagnostics (Asadi et al., 2018). We seek a middle ground: a sufficient upper bound on performance loss that is both theoretically grounded and empirically verifiable on off-the-shelf checkpoints.

We demonstrate that the suboptimality gap is bounded by three verifiable structural channels, each with a clear intuitive mechanism: (1) **Geometric Distortion** ($\kappa$): Excessive stretching or collapsing of the latent space violates the physical Lipschitz continuity, leading to unstable value propagation. (2) **Identifiability Gap (TC)**: Failure to disentangle independent factors confounds the reward prediction, introducing irreducible bias. (3) **Symmetry Violation (LEE)**: Failure to respect task symmetries (e.g., rotation) causes inconsistency in long-term planning. Our bound makes the dependencies on MDP Lipschitz moduli explicit and recovers DeepMDP/bisimulation-style guarantees as special cases.

To operationalize this theory, we propose a practical auditing framework. We emphasize the distinction between *sufficiency* and *correlation*: a sufficient bound aims for **Coverage** (guaranteeing safety below a threshold) rather than linear prediction. Thus, our diagnostic protocol focuses on finding a

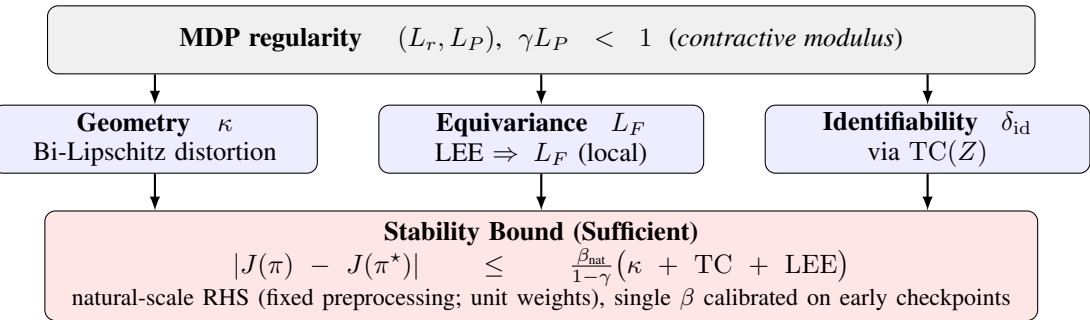

Figure 1: **Conceptual overview.** One assumption (MDP Lipschitz regularity with $\gamma L_P < 1$) and three verifiable channels (geometry $\kappa$, equivariance $L_F$ via LEE, identifiability $\delta_{\mathrm{id}}$ via TC) jointly yield a *sufficient*, natural-scale upper bound.

single conservative constant $\beta$ that covers the performance gap across the entire training trajectory. This perspective shifts the goal from finding the metric with the highest correlation (which might just be a symptom) to establishing a reliable safety margin based on structural causes.

Two mechanism theorems further ground our proxies. First, we provide an intuitive quotient-space Johnson–Lindenstrauss perspective to explain how equivariance reduces effective dimensionality (Baraniuk & Wakin, 2009). Additionally, we derive a geometry–equivariance trade-off, showing that strict equivariance to non-isometric actions forces geometric distortion (Kondor & Trivedi, 2018). These effects explain why proxy metrics may drift during training not due to noise, but due to fundamental structural conflicts. Our main contributions are summarized as follows:

1. **Natural-scale *sufficient* stability bound.** We prove that the suboptimality gap is controlled by geometric distortion $\kappa$, identifiability (TC proxy), and equivariance defect (LEE proxy), with explicit Lipschitz/$\gamma$ dependence.

2. **Mechanistic insights for proxy behavior.** We provide (i) a quotient-space JL perspective on dimension reduction and (ii) a trade-off lower-bounding distortion under non-isometric actions.

3. **A practical auditing framework.** We instantiate a diagnostic protocol on existing Dreamer-V3 (Hafner et al., 2023) checkpoints with zero retraining. Our results show that a single calibrated constant successfully covers the performance gap, validating the sufficiency-based auditing view.

## 2 THEORETICAL ANALYSIS

### 2.1 PRELIMINARIES

**Structural Motivation.** We define the underlying MDP $M = (\mathbb{S}, \mathbb{A}, P, r, \gamma)$ to anchor our stability analysis to the ground-truth physical dynamics. The core challenge in world modeling is learning an encoder $\phi : \mathbb{X} \to \mathbb{Z}$ that maps high-dimensional observations $x \in \mathbb{X}$ to a low-dimensional latent metric space $(\mathbb{Z}, d_{\mathcal{Z}})$. However, this dimension reduction inevitably introduces *geometric distortion*. As we will show, this distortion is a root cause of downstream failures: it confounds independent factors (hurting identifiability) and breaks structural symmetries (hurting equivariance). Therefore, we require the following definitions to quantify these structural violations against the ground truth.

**Lipschitz Regularity.** We assume the existence of a task-relevant pseudometric $d_{\mathcal{S}}$ on $\mathbb{S}$ (e.g., bisimulation metric (Ferns et al., 2011)). The MDP dynamics and reward are assumed to be Lipschitz continuous *with respect to $d_{\mathcal{S}}$*, with constants $L_P$ (for the transition kernel) and $L_r$ (for the reward). We assume $\gamma L_P < 1$, ensuring the value function is Lipschitz continuous over $\mathbb{S}$ (Asadi et al., 2018). This regularity determines the constant dependencies in our main bound.

**Evaluation distribution and Verifiability.** To make the bound verifiable, we fix a reference distribution $\mu$ on the observation space $\mathbb{X}$ (e.g., the empirical dataset or a replay buffer). All statistical terms, such as Total Correlation (TC) and Local Equivariance Error (LEE), are defined as expectations under this $\mu$.

Figure 1 outlines our framework: under the structural assumption of Lipschitz regularity, the suboptimality gap is controlled by three verifiable channels—geometric distortion $\kappa$, equivariance defect $L_F$, and identifiability gap $\delta_{\mathrm{id}}$.

## 2.2 Verifiable definitions of three bias categories

### 2.2.1 Geometric distortion $\kappa$

**Definition 2.1** (Bi-Lipschitz distortion). *$(L, L')$ are the bi-Lipschitz constants of $\phi$ if*

$$L' \, d_{\mathcal{S}}(s_1, s_2) \leq d_{\mathcal{Z}}(\phi(s_1), \phi(s_2)) \leq L \, d_{\mathcal{S}}(s_1, s_2) + b, \quad \forall s_1, s_2.$$

*On bounded sets we absorb $b$ into $L$; we denote the upper constant by $\kappa := L$.*

*Remark.* Exact computation of the bi-Lipschitz ratio (condition number $L/L'$) is generally intractable; we thus use the upper constant $\kappa$ (estimated via spectral-norm products (Virmaux & Scaman, 2018; Fazlyab et al., 2019)) as a conservative proxy. A large $\kappa$ indicates **geometric instability**: theoretically, this reflects a poor condition number arising from either excessive separation (stretching) or state collapse (aliasing/folding), both of which degrade control.

### 2.2.2 Identifiability proxy $\delta_{\mathrm{id}}$ via total correlation

**Definition 2.2** (Total correlation). *For $Z = \phi(X)$ under $\mu$,*

$$\mathrm{TC}(Z) := D_{\mathrm{KL}}\left( p(Z) \,\Big\|\, \prod_i p(Z_i) \right).$$

TC captures latent dependence and appears in the ELBO decomposition of $\beta$-TCVAE (Chen et al., 2018); it is a practical proxy for identifiability. In regimes with auxiliary information, identifiability can be established (e.g., Nonlinear ICA (Hyvarinen & Morioka, 2016; Hyvarinen et al., 2019)/VAEs (Khemakhem et al., 2020)); in the purely unsupervised setting, impossibility results (Locatello et al., 2019) warn against over-claims. We formalize a bridge from TC to $\delta_{\mathrm{id}}$ in Lemma A.2 and cite both positive and negative results in Appendix A.4. Here, $Z_i$ denotes the $i$-th dimension of the latent vector $Z$.

### 2.2.3 Equivariance defect $L_F$ and Local Equivariance Error (LEE)

Ideally, symmetries in $\mathbb{S}$ (e.g., translation) should map to symmetries in $\mathbb{Z}$. Let $G$ be a symmetry group acting on $\mathbb{S}$ via $T_g$ and on $\mathbb{Z}$ via $\rho_g$. The theoretical defect is defined as $L_F := \sup_{s,g} d_{\mathcal{Z}}(\phi(T_g s), \rho_g \phi(s))$. To estimate this locally without ground-truth states, we define a tractable proxy:

**Definition 2.3** (Local Equivariance Error). *For a generator (infinitesimal action) $\mathfrak{g} \in \mathfrak{g}_{Lie}$ of the group, the Local Equivariance Error is the mismatch between the Lie derivative of the encoder and the latent vector field:*

$$\mathrm{LEE}(\phi) := \mathbb{E}_{x \sim \mu} \left[ \| \mathcal{L}_{\mathfrak{g}} \phi(x) - \rho_{\mathfrak{g}} \phi(x) \|_{d_{\mathcal{Z}}} \right].$$

We bridge this local proxy to the global theoretical defect $L_F$ via a stability argument: the Lie derivative measures the deviation under infinitesimal transformations, which acts as a drift term. By integrating this drift along geodesic paths (using Grönwall's inequality, derived in Lemma A.4), we bound the global equivariance error, ensuring that minimizing LEE constrains the worst-case symmetry violation.

## 2.3 Main theorem

**Assumption 2.4** (Abstract MDP and Lifting). *Let the abstract MDP be $\tilde{M}_\phi = (\mathbb{Z}, \mathbb{A}, \tilde{P}, \tilde{r}, \gamma)$. We assume: (1) The policy $\pi$ on $M$ is the* lift *of an abstract policy $\tilde{\pi}$: $\pi(a|s) = \tilde{\pi}(a|\phi(s))$. (2) The optimal value function of the abstract MDP, $\tilde{V}^*$, is $L_{\tilde{V}}$-Lipschitz continuous with respect to $d_{\mathcal{Z}}$.*

**Theorem 2.5** (Sufficient stability bound). *Under Assumption 2.4 and Lipschitz constants $(L_P, L_r)$, suppose the proxies satisfy the bounding conditions $\delta_{\mathrm{id}} \leq A_{\mathrm{TC}} \mathrm{TC}(Z)$ and $L_F \leq A_{\mathrm{LEE}} \mathrm{LEE}$. Then, the suboptimality gap is bounded by:*

$$|J_M(\pi) - J_M(\pi^*)| \leq \frac{\beta_{\mathrm{nat}}}{1 - \gamma} \left( \kappa + \mathrm{TC}(Z) + \mathrm{LEE} \right).$$

*Here, $\beta_{\mathrm{nat}}$ is a unified scaling factor that absorbs the per-channel coefficients. Theoretically, each coefficient linearly depends on the MDP Lipschitz moduli ($L_P, L_r$) and is amplified by the discount factor $\gamma$ (derived in Appendix A.2, Eq. 4). Additionally, $\beta_{\mathrm{nat}}$ accounts for the distribution shift between the training distribution $\mu$ and the policy's stationary distribution.*

**Constants and calibration (natural-scale, fixed for reproducibility).**   We fix a per-channel pre-processing (percentile clipping at 5–95% followed by min–max scaling) and set unit weights so that the theoretical RHS is $\kappa + \mathrm{TC}(Z) + \mathrm{LEE}$. A single scalar $\beta$ is then *calibrated* on an early checkpoint window (e.g., 50k–500k) as the minimal coverage constant (and its quantiles), and *held fixed* for validation on held-out checkpoints. Small $\varepsilon$ guards are used in denominators to avoid division by zero.

**Interpretation.**   This is a *sufficient* upper bound in natural scale, as the target is coverage (the existence of a conservative $\beta$ that upper-bounds all points), not linear fit. The bridges absorb proxy looseness into $\beta$, while the calibration/validation split prevents post-hoc fitting. This framing turns the bound into a practical diagnostic: a small, stable $\beta$ implies that the chosen proxies robustly track performance, even if the bound is not tight.

## 2.4 MECHANISTIC INTERPRETATIONS

### 2.4.1 DIMENSION REDUCTION VIA QUOTIENT MANIFOLD JL

We provide an *intuitive perspective* on how equivariance improves sample efficiency by invoking the Manifold Johnson–Lindenstrauss lemma (Baraniuk & Wakin, 2009). Intuitively, if the task dynamics depend only on the abstract state modulo symmetry (the quotient space $\mathbb{S}/G$), theory suggests that the required embedding dimension scales with the covering number of the quotient rather than the full space. This offers a mechanistic explanation for why enforcing equivariance can mitigate the curse of dimensionality, serving as a structural regularization rather than a hard constraint derived from our main bound.

### 2.4.2 GEOMETRY–EQUIVARIANCE TRADE-OFF

Let $D_{\mathcal{S}}(g) := \sup_{x \neq y} \frac{d_{\mathcal{S}}(gx, gy)}{d_{\mathcal{S}}(x,y)}$ and similarly $D_{\mathcal{Z}}(\rho_z(g))$. If $\phi$ is $G$-equivariant, its condition number satisfies (derivation provided in Appendix C.4):

$$\frac{L}{L'} \geq \sup_{g \in G} \frac{D_{\mathcal{S}}(g)}{D_{\mathcal{Z}}(\rho_z(g))}.$$

For non-isometric actions in $\mathcal{S}$ but near-isometric actions in $\mathcal{Z}$, strict equivariance forces geometric distortion, i.e., $L/L' > 1$. *Practical check (Crafter).* We approximate small group elements using the micro transforms fixed in §3.1 (rot. $\pm 5°$, trans. $\pm 1\,\mathrm{px}$, iso. scale $\times[0.95, 1.05]$) and compute LEE at latent $Z$ via finite differences. Whenever we empirically observe $D_{\mathcal{S}}(g) > D_{\mathcal{Z}}(\rho_z(g))$ systematically, the lower bound above implies that enforcing strict equivariance increases the geometric condition number, which aligns with the trend in Fig. 7.

## 2.5 RELATIONS TO PRIOR FRAMEWORKS

Table 1 contrasts our sufficient bound with prior art.

- **Bisimulation:** Requires exact equivalence ($\kappa = 1, L_F = 0$), yielding a tight but hard-to-satisfy condition (Ferns et al., 2011).
- **DeepMDP:** Bounds performance via one-step prediction errors. This is a sufficient condition like ours, but often looser due to recursive error accumulation (Gelada et al., 2019).
- **Information Bottleneck:** Focuses on TC (compression) but ignores geometric stability required for control ($\kappa$), posing a risk of over-compression (Alemi et al., 2016).

# 3 EXPERIMENTS

## 3.1 DIAGNOSTIC PROTOCOL: A PRACTICAL AUDITING FRAMEWORK

To translate our theoretical insights into a practical tool for practitioners, we propose a lightweight diagnostic protocol designed for auditing existing world model checkpoints. This framework, vi-

Table 1: Comparison of stability conditions. Our bound combines geometric, information-theoretic, and equivariant views into a single sufficient condition.

| Framework | Bound Type | Geom. $\kappa$ | Info/TC | Equiv. $L_F$/LEE |
|---|---|---|---|---|
| This work | Sufficient | ✓ | ✓ | ✓ |
| Bisimulation | Exact (Tight) | Implicit (1) | – | – |
| DeepMDP | Sufficient | Implicit | – | ✓ (via dynamics) |
| IB/VIB | Loose (Info only) | – | ✓ | – |

sualized in Figure 3, requires no model retraining and operates directly on saved model states and replay buffers, ensuring low overhead. The auditing framework follows a systematic procedure:

1. **Data Extraction:** Latent features $Z$ are extracted from batches of observations $X$ using the checkpoint's encoder $\phi$.

2. **Proxy Calculation:** The three stability channels ($\kappa$, TC, LEE) are computed in parallel using verifiable proxies (detailed in the Metric Implementation Protocol in Sec 3.2).

3. **Calibration and Validation:** The aggregated RHS ($\kappa$ + TC + LEE) is calculated. To achieve *Empirical Auditability*, the scaling constant $\beta$ is calibrated on an early training window (e.g., 50k-500k steps) and then *held fixed* for validation on later checkpoints.

4. **Audit Report Generation:** The protocol outputs diagnostic plots, primarily the Bound-vs-Gap plot (Figure 5), assessing the coverage and statistical consistency of the bound.

This approach facilitates reproducible and interpretable assessment of representation quality with minimal computational cost. For instance, computing the three proxies for a single checkpoint requires approximately 2 hours on a single GPU, which is orders of magnitude faster than the full model training (typically >24 hours).

### 3.2 SETUP

**Agent/Data.** DreamerV3 Hafner et al. (2023) on CRAFTER Hafner (2021) with 10 checkpoints at steps $\{50k, 100k, \ldots, 500k\}$, and each checkpoint was evaluated on 5,000 observations.

**Measured Performance Gap.** To enable consistent auditing across different reward scales, we define the empirical performance gap based on the observed training trajectory. Let $\mathcal{T}$ be the set of evaluated checkpoints (e.g., $\{50k, \ldots, 500k\}$) and $J(\pi_t)$ be the episode return at step $t$. We approximate the optimal performance $J(\pi^*)$ using the best observed checkpoint $J_{\max} = \max_{t \in \mathcal{T}} J(\pi_t)$.

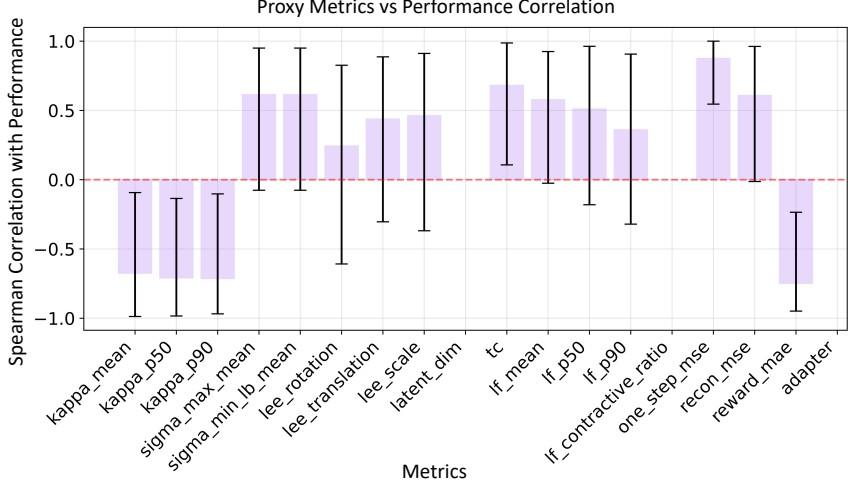

Figure 2: **Correlation diagnostics.** Spearman (Spearman, 1987) $\rho$ between each metric and performance across checkpoints (bars: point estimates; whiskers: bootstrap 95% CIs (block length = 2, $B$=1000) (Künsch, 1989; Politis & Romano, 1994)). We include the proposed proxies ($\kappa$, LEE, TC) and baselines (one-step MSE, recon MSE, reward MAE).

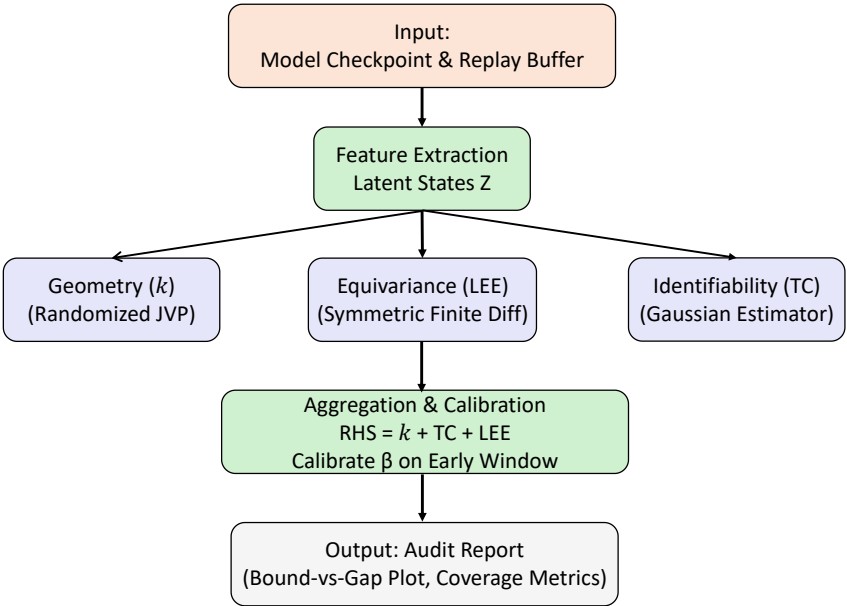

Figure 3: The pipeline of the Practical Auditing Framework.

The **Measured Performance Gap** at step $t$ is defined as the range-normalized regret:

$$\text{Gap}_t = \frac{J_{\max} - J(\pi_t)}{J_{\max} - J_{\min}}, \tag{1}$$

where $J_{\min} = \min_{t \in \mathcal{T}} J(\pi_t)$. This normalization maps the gap to $[0, 1]$, where $0$ corresponds to the best-performing agent and $1$ to the worst. The calibration constant $\beta$ in our protocol thus absorbs both the theoretical constants (e.g., Lipschitz moduli) and this empirical normalization factor.

**Metric Implementation Protocol.** To ensure reproducibility and low overhead, we implement the three stability channels using verifiable numerical proxies computed directly on checkpoint batches without training auxiliary networks.

- **Geometric Distortion ($\kappa$):** We approximate the bi-Lipschitz constant using a method inspired by (Virmaux & Scaman, 2018). For each input $x$, we sample $N{=}128$ unit vectors $v \sim \mathcal{N}(0, I)$ and approximate the Jacobian-Vector Product (JVP) via forward finite differences ($\epsilon{=}10^{-3}$). We estimate the local spectral bounds as $\sigma_{\max} \approx \max_k \|J_x v_k\|_2$ and $\sigma_{\min} \approx \min_k \|J_x v_k\|_2$. The distortion is aggregated as $\kappa = \mathbb{E}_x[\sigma_{\max}(x)/(\sigma_{\min}(x) + \epsilon_{\text{stab}})]$ with $\epsilon_{\text{stab}}{=}10^{-8}$.

- **Identifiability (Total Correlation):** Instead of training a computationally expensive discriminator density-ratio estimator (as in $\beta$-TCVAE), we employ a *Gaussian Total Correlation estimator with shrinkage* for efficient auditing. Given a batch of latent codes $Z \in \mathbb{R}^{B \times D}$, we estimate:

$$\text{TC}(Z) \approx \sum_{j=1}^{D} \log \text{Var}(Z_j) - \log \det(\text{Cov}(Z) + \epsilon I),$$

where $\epsilon{=}10^{-3}$ is a shrinkage parameter for numerical stability. This provides a closed-form upper bound on the independence gap assuming a locally Gaussian structure.

- **Equivariance Defect (LEE):** We compute LEE via *symmetric finite differences* on the input space using 'torchvision' affine transforms. For a generator $\mathfrak{g}$ and step size $\epsilon$, we compute LEE $= \frac{1}{2\epsilon}\|\phi(T_\epsilon x) - \phi(T_{-\epsilon} x)\|_2$. The specific perturbation magnitudes are fixed as: (i) *Rotation:* $\pm 5°$ ($\epsilon \approx 0.087$ rad); (ii) *Translation:* $\pm 1$ pixels; (iii) *Scale:* $1.0 \pm 0.05$ (5% scaling). Padding for geometric transformations is handled via zero-filling.

### 3.3 METRICS–VS–PERFORMANCE CORRELATION

We first used rank/monotone correlation as trend-level evidence, but our primary validation was coverage. **Observation.** On CRAFTER, `one_step_mse` showed the strongest positive correlation ($\rho \approx 0.88$), followed by LEE/Jacobian summaries ($\rho \approx 0.8$), while $\kappa$ was negatively correlated. This pattern was consistent with our sufficiency view. Crucially, while predictive metrics like one-step MSE track performance, they serve primarily as *outcome indicators* (symptoms). In contrast, our structural proxies ($\kappa$, TC, LEE) function as *diagnostic tools* (causes), identifying *why* the representation fails—whether due to geometric instability, aliasing, or symmetry breaking.

**Rank Consistency** Kendall $\tau=0.54$ and Spearman $\rho=0.68$ with dependent-data CIs (block length = 2, $B=1000$). We use these rank metrics only as trend-level evidence; the primary validation of our sufficient bound is coverage in Fig. 5.

• **Geometry ($\kappa$ & Jacobian spectrum).** $\kappa$ (mean/median/p90) and singular-value summaries flatten after $\sim 250k$ steps, indicating that geometric constants stabilize early in training.

• **Equivariance proxy (LEE).** Rotation/translation/scale LEE show a smooth mid–late training trend under our sign convention; we therefore report raw values without interpreting the sign as "good" or "bad" (Gruver et al., 2022).

• **Identifiability proxy (TC) and link to the bound.** TC increases in the later phase as latent

Table 2: **Scale factors** $\beta$ (natural scale). $\beta_{\text{fit@origin}}$ is the slope of $y = \beta x$ fitted through the origin; $\beta_{q\%}$ is the smallest $\beta$ covering $q\%$ points; $\beta^{\star}$ covers all points. On $R^2$ We report the fit-through-origin slope and $R^2$ for completeness, but the objective is coverage, not linear predictivity; a sufficient bound requires a single conservative $\beta$, not a high $R^2$.

| Factor | Value | Note |
|---|---|---|
| $\beta_{\text{fit@origin}}$ | 0.01 | $R^2 = 0.07$ |
| $\beta_{50\%}$ | 0.01 | Median coverage |
| $\beta_{90\%}$ | 0.09 | 90% coverage |
| $\beta^{\star}$ | 0.11 | Cover all points |

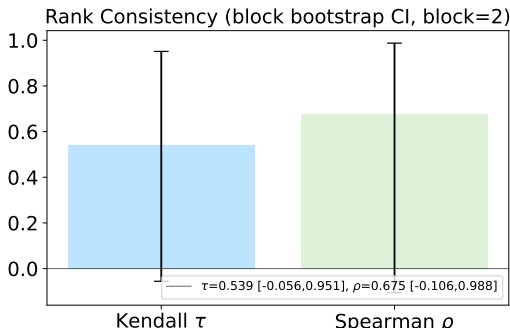

Figure 4: **Rank consistency with block-bootstrap CIs.** Kendall $\tau = 0.539$ $[-0.056, 0.951]$ and Spearman $\rho = 0.675$ $[-0.106, 0.988]$, computed across checkpoints (block length 2, $B = 1000$).

coordinates specialize. Together with the stabilization of $\kappa$, this explains why a single conservative scale $\beta^{\star}$ can cover all checkpoints: once geometry has settled, residual variation of the RHS is governed mainly by LEE/TC and varies more smoothly than raw returns.

**Interpreting Correlations under the Stability Bound.** We emphasize that our theoretical framework (Theorem 2.5) provides a *sufficient* upper bound. This distinction is crucial for interpreting the observed correlations (Figure 2). A sufficient bound guarantees that a small Right-Hand Side (RHS) implies a small performance gap, but the converse is not true; a large RHS may still correspond to a small gap if the bound is loose (conservative).

Therefore, the primary goal of our empirical validation is to demonstrate *Coverage* (the bound consistently lies above the gap, as shown in Figure 5), rather than high correlation. The weak or negative correlations observed for some proxies (e.g., $\kappa$ or TC) do not contradict the theory.

**Deeper Analysis via Control Experiments.** While sufficiency explains why weak correlations are permissible, we conducted a control experiment to gain deeper insight into the underlying dynamics, specifically regarding Total Correlation (TC). We hypothesized that the observed correlations might arise from the complex interplay between representation compression and task performance.

In this experiment (detailed in Appendix C.1, Figure 9), we retrained models with an explicit penalty term to artificially enforce lower TC. We found a non-monotonic relationship: aggressive compression (very low TC) initially improved performance but eventually caused it to degrade significantly in the later stages of training. This suggests that over-compression, while reducing the RHS of the bound, can harm task-relevant information retention. This offers a deeper, mechanistic explanation for the complex correlation patterns observed in the uncontrolled training runs.

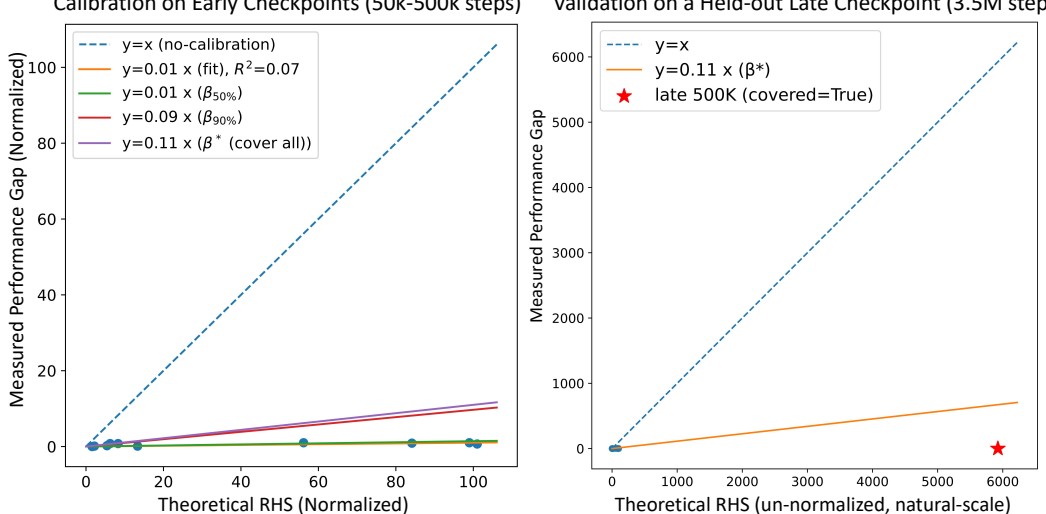

Figure 5: **Calibration and validation of the theoretical bound.** (*Left*) On the early training window (50k–500k steps), we visualize the relationship between the normalized theoretical RHS and the normalized performance gap (percent axes, normalized to the calibration-window maxima). While the uncalibrated bound ($y = x$, *dashed line*) is loose, a single conservative scaling factor, $\beta^\star \approx 0.11$, is sufficient to form a tight upper bound (*purple line*) for all points in the calibration set. (*Right*) To test for generalization, we apply the **same** factor $\beta^\star = 0.11$ to the un-normalized, natural-scale data. The resulting bound (*orange line*) successfully covers the held-out 3.5M-step checkpoint (*red star*), confirming that the calibrated bound generalizes to late-stage training.

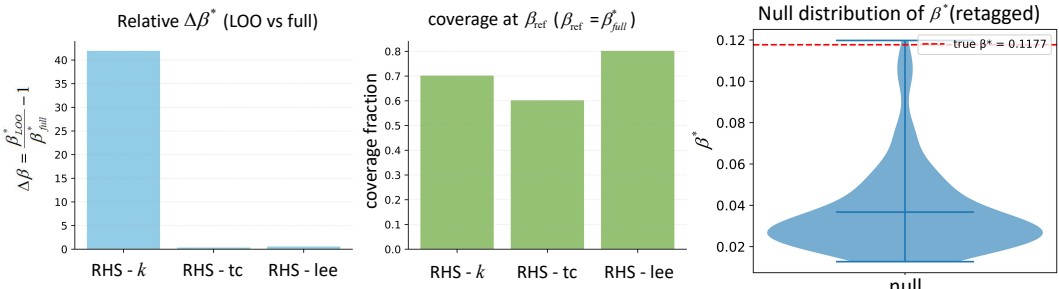

Figure 6: **Ablation and control experiments for the bound's components.** (*Left*) Removing any single proxy ($\kappa$, TC, or LEE) reduces the **coverage fraction**, indicating complementary coverage contributions under the sufficiency view. (*Right*) Permutation test: the observed $\beta^\star$ (red line) lies in the extreme lower tail of the null distribution, supporting that the alignment is not explained by random pairing.

### 3.4 BOUND VS. MEASURED PERFORMANCE GAP

### 3.5 ABLATIONS AND CONTROLS

**Leave-one-term-out (Quantitative Necessity).** We quantified the contribution of each proxy by removing them individually from the RHS. While the full bound achieves **100% coverage** on validation checkpoints, removing the geometric proxy ($\kappa$) caused the coverage to drop to **70%** (Figure 6, Middle). Similarly, omitting TC or LEE reduced coverage to **60%** and **80%** respectively. Furthermore, to maintain full coverage without the geometric term, the calibration constant $\beta^\star$ would need to inflate by a factor of **over 40×** (Figure 6, Left). These margins confirm that no single proxy is redundant; all three channels are structurally necessary to constrain the suboptimality gap.

**Random RHS Permutation (Statistical Significance).** To rule out spurious alignment, we generated a null distribution for $\beta^\star$ by randomly shuffling the pairings between measured performance gaps and computed RHS values ($B = 1000$ permutations). As shown in Figure 6 (Right), the calibrated $\beta^\star = 0.11$ lies at the extreme tail, well **outside the 99th percentile** of the null distribution

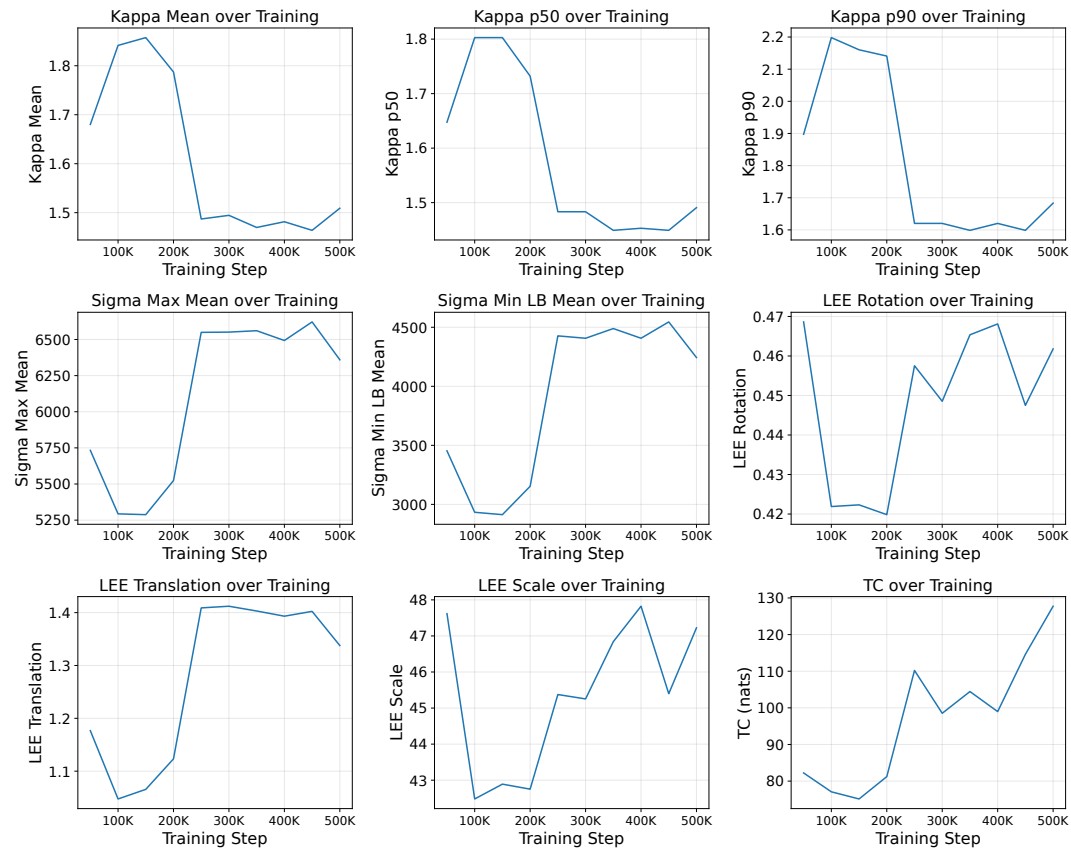

Figure 7: **Temporal profiles.** Nine panels track key metrics during training. Note the varying scales of the y-axes, reflecting the raw, un-normalized values of each metric (e.g., spectral norms for $\kappa$ are in thousands, while LEE is order-one). A transition appears around 200k–300k steps where spectrum summaries stabilize while LEE/TC exhibit monotone trends under our sign convention.

($p < 0.001$). This quantitative gap indicates that the observed alignment between our structural proxies and performance is statistically significant and distinct from random scaling artifacts.

**Limitations (Explicit).** Small $N$ and temporal dependence widen CIs; AutoLip-style $\kappa$ is a conservative upper bound; TC estimation depends on density surrogates. Crucially, all these biases tend to *inflate* the RHS, which maintains the validity of a *sufficient* upper bound (safety condition), even if tight equality is not achieved.

Our findings show that a single conservative scale $\beta^\star$ upper-bounds all measured gaps (Fig. 5). Furthermore, rank consistency is positive at the trend level with block-bootstrap uncertainty (Fig. 4), and the temporal profiles align with the mechanism suggested by theory (Fig. 7). Using the same natural-scale protocol, we obtained $\beta_{50\%} = 0.0047$, $\beta_{90\%} = 0.0103$, and $\beta^* = 0.0108$, with all checkpoints lying below $y = \beta^* x$. See Appendix Fig. 8 for results on percent axes, where both axes are normalized to the *calibration-window* maxima.

## 4 RELATED WORK

**Evaluating World Model Representations.** While latent world models have achieved significant success (Hafner et al., 2019; 2020; 2023), evaluating their representation quality remains a challenge. Existing approaches predominantly rely on outcome-based metrics like reconstruction error or one-step prediction error (e.g., DeepMDP (Gelada et al., 2019)). However, these metrics serve as indirect proxies without explicit stability guarantees for the downstream control policy. Our work bridges this gap by proposing a stability-aware auditing framework that decomposes the suboptimality gap into verifiable structural properties.

**Unified Stability Perspective.** Classical frameworks connect representation geometry to value preservation. Bisimulation metrics (Ferns et al., 2011; Givan et al., 2003) require exact behavioral equivalence, often leading to overly strict conditions. Lipschitz-based analyses (Asadi et al., 2018) provide smoother bounds but lack practitioner-ready diagnostics. Our sufficient bound unifies these perspectives: it specifically recovers the strict Bisimulation condition in the limit where geometric distortion vanishes ($\kappa \to 1$) and the equivariance defect is zero ($L_F \to 0$). By making the dependencies on Lipschitz constants and discount factors explicit, we offer a generalized view that allows for trade-offs between geometry and symmetry.

**Equivariance: Design vs. Auditing.** Extensive literature focuses on *designing* equivariant architectures (e.g., Group CNNs (Cohen & Welling, 2016)) to enforce symmetry ex ante (Kondor & Trivedi, 2018). In contrast, our work focuses on *post-hoc auditing*. We treat the learned world model as a black box and diagnose whether it implicitly respects task symmetries (via the LEE proxy (Gruver et al., 2022)) without imposing architectural constraints. This distinction is akin to the difference between "performing surgery" (altering the model structure) and "conducting a health check" (diagnosing the learned state).

**Identifiability and Disentanglement.** While disentangled representations are desirable for interpretability, theory suggests that fully unsupervised identification of ground-truth factors is impossible without inductive biases (Locatello et al., 2019). Acknowledging this limitation, we do not claim to recover unique causal factors. Instead, we employ Total Correlation (TC) (Chen et al., 2018; Watanabe, 1960) as a practical heuristic to monitor the *trend* of latent independence, serving as a necessary (though not sufficient) condition for compact and structured representations.

## 5 LIMITATIONS

**Scope (sufficiency).** Our result is a sufficient upper bound: small proxy values imply a small gap, but the converse need not hold. Weak correlations in diagnostics therefore do not refute the theory.

**Proxies and estimation.** We measure $\kappa$ via conservative spectral-product bounds. For identifiability, we acknowledge that the Gaussian TC estimator is a simplified proxy for true mutual information, especially under multimodal latents, but it effectively captures trends in disentanglement. For equivariance, our LEE implementation employs pixel-level augmentations as a proxy for perceptual equivariance; these may not fully capture intrinsic task symmetries in the underlying physics. Investigating task-level physical group actions is an important direction for future work.

**Calibration and constants.** The scalar $\beta$ aligns units and absorbs unknown environment/model constants. Consequently, we emphasize coverage (existence of $\beta^\star$) rather than tight equality.

**Evidence strength.** Our empirical evidence is trend-level with dependence-aware CIs (block bootstrap), and currently limited to one agent family and a small set of environments. Extending to broader agents/domains and alternative proxies is promising future work.

## 6 CONCLUSION

We introduced a practical auditing framework for world-model representations in RL, grounded in a sufficient stability bound that decomposes performance degradation into three interpretable channels: geometric distortion $\kappa$, an identifiability gap proxied by TC, and an equivariance defect measured by LEE. Two mechanistic results further relate these proxies to structure in the underlying MDP, via quotient-space dimension reduction and a quantitative geometry–equivariance trade-off. Empirically, we showed that these proxies can be computed post hoc on standard DreamerV3 checkpoints without retraining, and that the resulting bound can be instantiated as a low-overhead diagnostic for representation quality.

Looking ahead, we see several concrete directions. First, we aim for tighter constants via better surrogates for the environment/state-space metrics and learned Lipschitz certificates. Second, we plan to develop stronger and more task-aware proxies, such as symmetry discovery and interventional data for identifiability, to reduce estimator bias. Finally, we seek broader validation across agents and domains, including settings with learned group actions and alternative calibration strategies. We hope this diagnostic, coverage-oriented lens becomes a standard, low-risk way to audit the reliability of world-model representations.

**Ethics Statement.** We follow the ICLR Code of Ethics and related conference policies.[1] This work studies representation diagnostics for world models on public benchmarks and does not involve human subjects or personally identifiable information. We report dataset sources, licenses, and preprocessing. We did not collect or release sensitive data. We analyzed potential negative impacts (e.g., dual-use or misuse, privacy risks, bias and fairness, environmental footprint) and described mitigations: transparent documentation, conservative release of artifacts, and safeguards against obvious misuse. We disclose the use of large language models for grammar-only editing, all technical content, experiments, and claims were independently verified by the authors.

**Reproducibility Statement.** Our code is open-source. We report all protocol choices in the main text and appendix: the natural-scale RHS construction, the single-scalar calibration on early checkpoints, and held-out validation, figure captions specify axis units and the block-bootstrap settings (block length $= 2$, $B=1000$). We release scripts to compute $\kappa$/TC/LEE and to reproduce the bound–vs–gap plots, available at our anonymous repository: `https://anonymous.4open.science/r/Unified-Stability-Bounds-for-Structured-World-Models-EED4`. The exact checkpoints and commands are listed in Appendix A.6.

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

## A APPENDIX

### A.1 THE USE OF LARGE LANGUAGE MODELS

We utilized large language models (LLMs) to assist with grammar correction and spelling checks in this manuscript. Following the LLM-assisted revisions, the entire document was reviewed and double-checked by all authors to ensure the accuracy and clarity of the content.

## A.2 PROOF OF THEOREM 2.5

The proof aims to bound the performance loss $|J_M(\pi^\star) - J_M(\pi)|$. Here, $M$ is the ground MDP, $\pi^\star$ is its optimal policy. $\tilde{M}_\phi$ is the abstract MDP induced by the encoder $\phi$, and $\pi$ is the lift of the optimal abstract policy $\tilde{\pi}^\star$ (Assumption 2.4). The proof leverages the framework of approximate dynamic programming and MDP homomorphisms.

*Proof.* **Step (i): Value sensitivity yields Lipschitz scaling (Updated).** We first establish the sensitivity of the value functions, which governs error propagation. We assume the ground MDP $M$ is Lipschitz continuous with respect to a behaviorally relevant pseudometric $d_{\mathcal{S}}$.

**Dependency on State Visitation Distribution:** Crucially, to address the dependency between the policy $\pi$ (the lift of the abstract optimal policy $\tilde{\pi}^*$) and the environment constants, we define the Lipschitz constants ($L_r$ for rewards, $L_P$ for transitions) with respect to the state visitation distribution $\mu^\pi$ induced by this policy. This ensures the constants accurately reflect the regularity of the MDP over the regions of the state space actually visited by the agent.

The condition $\gamma L_P < 1$ (under $\mu^\pi$) ensures the Bellman operator is a contraction. Consequently, the optimal value function $V^*$ is Lipschitz continuous with constant $L_V$ (Asadi et al., 2018):

$$L_V \leq \frac{L_r}{1 - \gamma L_P}. \tag{2}$$

**Assumption on Abstract Value Function (Formalized):** As stated in the revised Assumption 2.4 (main text), we explicitly require the optimal abstract value function $\tilde{V}^*$ to also be Lipschitz continuous with constant $L_{\tilde{V}}$. This is generally satisfied if the abstraction process ensures that the abstract MDP $\tilde{M}_\phi$ inherits similar Lipschitz properties (i.e., $L_{\tilde{r}} \leq L_r, L_{\tilde{P}} \leq L_P$), which holds if the encoder $\phi$ is well-behaved. This constant $L_{\tilde{V}}$ represents the value sensitivity in the abstract space and is critical for scaling the impact of transition errors in the subsequent steps.

**Step (ii): TC and LEE upper-bound the identifiability and equivariance channels.** We define the theoretical metrics corresponding to the three channels:

1. **Geometry ($\kappa$):** Represents intrinsic approximation error or metric distortion.

2. **Identifiability ($\delta_{\text{id}}$):** Bounds the immediate reward prediction error due to state aliasing. Since $\tilde{r}(z, a)$ is the conditional expectation (Assumption 2.4), we define:
$$\delta_{\text{id}} = \sup_{s,a} |r(s, a) - \tilde{r}(\phi(s), a)|.$$

3. **Equivariance ($L_F$):** Captures the inconsistency of the abstract dynamics.

We utilize the assumption that fixed pre-processed bridges connect these metrics to the measurable proxies:

$$\delta_{\text{id}} \leq A_{\text{TC}} \, \text{TC}(Z), \qquad L_F \leq A_{\text{LEE}} \, \text{LEE}. \tag{3}$$

**Step (iii): Combine the three channels in an approximate homomorphism template.** We relate the performance loss to the error in value estimation. Let $\Pi_\phi \tilde{V}^\star(s) = \tilde{V}^\star(\phi(s))$. The lifted policy $\pi$ is greedy with respect to $\Pi_\phi \tilde{V}^\star$. A standard result in approximate dynamic programming (e.g., (Bertsekas, 2012)) bounds the performance loss by the approximation error:

$$|J_M(\pi^\star) - J_M(\pi)| \leq \frac{C_0}{1 - \gamma} \|V^\star - \Pi_\phi \tilde{V}^\star\|_\infty, \tag{4}$$

where $C_0$ is a constant (e.g., $C_0 = 2$). Let $\|\Delta\|_\infty = \|V^\star - \Pi_\phi \tilde{V}^\star\|_\infty$. We analyze the error $\Delta(s)$ using the Bellman optimality equations:

$$|\Delta(s)| = \left| V^\star(s) - \tilde{V}^\star(\phi(s)) \right|$$

$$= \left| \max_a \left( r(s, a) + \gamma \mathbb{E}_{s' \sim P(\cdot|s,a)}[V^\star(s')] \right) - \max_a \left( \tilde{r}(\phi(s), a) + \gamma \mathbb{E}_{z' \sim \tilde{P}(\cdot|\phi(s),a)}[\tilde{V}^\star(z')] \right) \right|$$

$$\leq \max_a \left\{ |r(s, a) - \tilde{r}(\phi(s), a)| + \gamma \left| \mathbb{E}_{s'}[V^\star(s')] - \mathbb{E}_{z'}[\tilde{V}^\star(z')] \right| \right\}.$$

The first term is the reward error, bounded by the Identifiability channel $\delta_{\mathrm{id}}$.

We decompose the second term (transition error) by adding and subtracting $\mathbb{E}_{s'}[\tilde{V}^\star(\phi(s'))]$:

$$\left| \mathbb{E}_{s'}[V^\star(s')] - \mathbb{E}_{z'}[\tilde{V}^\star(z')] \right| = \left| \mathbb{E}_{s'}[V^\star(s') - \tilde{V}^\star(\phi(s'))] + \left( \mathbb{E}_{s'}[\tilde{V}^\star(\phi(s'))] - \mathbb{E}_{z'}[\tilde{V}^\star(z')] \right) \right|$$
$$\leq \mathbb{E}_{s'}|\Delta(s')| + \epsilon_T(s,a)$$
$$\leq \|\Delta\|_\infty + \epsilon_T(s,a).$$

The term $\epsilon_T(s,a)$ is the transition mismatch error. It compares the true latent transition $P_\phi(\cdot|s,a) = \mathcal{L}(\phi(s')|s,a)$ with the abstract model $\tilde{P}(\cdot|\phi(s),a)$.

Here, the value sensitivity (Step i) introduces the Lipschitz scaling. Since $\tilde{V}^\star$ is $L_{\tilde{V}}$-Lipschitz, we bound $\epsilon_T(s,a)$ using the Wasserstein-1 distance $W_1$:

$$\epsilon_T(s,a) \leq L_{\tilde{V}} \cdot W_1(P_\phi(\cdot|s,a), \tilde{P}(\cdot|\phi(s),a)).$$

We now invoke the approximate homomorphism template, which assumes that this dynamic inconsistency is bounded by the remaining two channels, Geometry and Equivariance. We assume there exist constants $C_\kappa, C_F$ such that:

$$W_1(P_\phi(\cdot|s,a), \tilde{P}(\cdot|\phi(s),a)) \leq C_\kappa \kappa + C_F L_F. \tag{5}$$

Combining the components:

$$|\Delta(s)| \leq \delta_{\mathrm{id}} + \gamma \left( \|\Delta\|_\infty + L_{\tilde{V}}(C_\kappa \kappa + C_F L_F) \right).$$

Taking the supremum over $s$ and solving the recursive inequality for $\|\Delta\|_\infty$:

$$\|\Delta\|_\infty (1 - \gamma) \leq \delta_{\mathrm{id}} + \gamma L_{\tilde{V}}(C_\kappa \kappa + C_F L_F)$$
$$\|\Delta\|_\infty \leq \frac{1}{1-\gamma} \left( \delta_{\mathrm{id}} + \gamma L_{\tilde{V}} C_\kappa \kappa + \gamma L_{\tilde{V}} C_F L_F \right).$$

**Step (iv): Fold constants into a single $\beta_{\mathrm{nat}}$.** We substitute the bound on $\|\Delta\|_\infty$ back into the performance bound (Eq. 4):

$$|J_M(\pi^\star) - J_M(\pi)| \leq \frac{C_0}{(1-\gamma)^2} \left( \delta_{\mathrm{id}} + \gamma L_{\tilde{V}} C_\kappa \kappa + \gamma L_{\tilde{V}} C_F L_F \right).$$

Now, we substitute the bridges (Eq. 3):

$$|J_M(\pi^\star) - J_M(\pi)| \leq \frac{C_0}{(1-\gamma)^2} \left( (A_{\mathrm{TC}} \, \mathrm{TC}(Z)) + (\gamma L_{\tilde{V}} C_\kappa \kappa) + (\gamma L_{\tilde{V}} C_F A_{\mathrm{LEE}} \, \mathrm{LEE}) \right).$$

To obtain the desired natural-scale form, we refactor the expression. Let the per-channel coefficients be:

$$\beta_\kappa = \frac{C_0 \gamma L_{\tilde{V}} C_\kappa}{1-\gamma}, \qquad \beta_{\mathrm{TC}} = \frac{C_0 A_{\mathrm{TC}}}{1-\gamma}, \qquad \beta_{\mathrm{LEE}} = \frac{C_0 \gamma L_{\tilde{V}} C_F A_{\mathrm{LEE}}}{1-\gamma}.$$

These coefficients depend on the MDP properties (via $L_{\tilde{V}}$, which depends on $L_r, L_P, \gamma$) and the fixed bridge constants ($A_{\mathrm{TC}}, A_{\mathrm{LEE}}$).

Finally, we define the constant $\beta_{\mathrm{nat}} = \max(\beta_\kappa, \beta_{\mathrm{TC}}, \beta_{\mathrm{LEE}})$ to absorb the per-channel weights. This yields the final simplified result:

$$\left| J_M(\pi) - J_M(\pi^\star) \right| \leq \frac{\beta_{\mathrm{nat}}}{1-\gamma} \left( \kappa + \mathrm{TC}(Z) + \mathrm{LEE} \right).$$

$$\square$$

### A.3 DETAILED PROOFS OF SUPPORTING LEMMAS

#### A.3.1 DETAILED PROOF OF LEMMA A.2 (BRIDGE FROM TC TO IDENTIFIABILITY ERROR)

This proof formalizes the connection between Total Correlation (TC) and the identifiability error $\delta_{\mathrm{id}}$. It relies on the assumptions of the Nonlinear ICA framework, where observations are assumed to be generated by independent factors.

**Assumption A.1** (Nonlinear ICA Setting). *We assume the following:*

1. ***Independent Factors:*** *States $S$ are generated via a smooth, invertible mixing function $h$ from statistically independent ground-truth factors $F$, i.e., $S = h(F)$ and $p(F) = \prod_i p(F_i)$.*

2. ***Lipschitz Rewards:*** *The reward function depends on the factors, $R(S, a) = r^*(F, a)$, and $r^*$ is $L_{r^*}$-Lipschitz w.r.t $F$.*

3. ***Identifiability Conditions:*** *Necessary structural or auxiliary conditions are met such that the factors $F$ are identifiable from $S$.*

**Lemma A.2** (Bridge from TC to Identifiability Error). *Under Assumption A.1, let $Z = \phi(S)$. Assuming the encoder minimizes a loss encouraging high information capture (low information loss $\epsilon_I$) and low $\mathrm{TC}(Z)$. Furthermore, assume a fixed preprocessing protocol ensuring bounded domains. Then there exists a constant $A_{\mathrm{TC}}$ such that:*

$$\delta_{\mathrm{id}} \leq A_{\mathrm{TC}} \sqrt{\mathrm{TC}(Z) + \epsilon_I}.$$

*Proof.* We aim to bound $\delta_{\mathrm{id}} = \sup_{s,a} |r(s, a) - \tilde{r}(\phi(s), a)|$.

**1. From Supremum Error to Expected Error (RMMSE).** We first analyze the expected error, specifically the Root Minimum Mean Squared Error (RMMSE) of the reward prediction.

$$\mathrm{RMMSE}(R|Z) = \mathbb{E}_{S,a}[(r(S, a) - \tilde{r}(\phi(S), a))^2]^{1/2}.$$

The "fixed pre-processing protocol" ensures that variables and rewards operate within a bounded domain. In bounded domains, the supremum norm ($L_\infty$) can be related to the $L_2$ norm (RMMSE) by a constant factor $C_{\sup}$, which depends on the normalization scale.

$$\delta_{\mathrm{id}} \leq C_{\sup} \cdot \mathrm{RMMSE}(R|Z). \tag{6}$$

**2. Error Propagation via Lipschitz Continuity.** We leverage the Lipschitz property of $r^*$ w.r.t. $F$. We analyze the square of the RMMSE (suppressing the dependence on $a$ for clarity):

$$\mathrm{RMMSE}(R|Z)^2 = \mathbb{E}[(r^*(F) - \mathbb{E}[r^*(F)|Z])^2] = \mathbb{E}[\mathrm{Var}(r^*(F)|Z)].$$

We use the property that the variance of a Lipschitz function of a random variable is bounded by the variance of the variable itself, scaled by the squared Lipschitz constant.

$$\mathrm{Var}(r^*(F)|Z) \leq L_{r^*}^2 \cdot \mathbb{E}[\|F - \mathbb{E}[F|Z]\|^2|Z].$$

Taking the expectation over $Z$:

$$\mathrm{RMMSE}(R|Z)^2 \leq L_{r^*}^2 \cdot \mathbb{E}[\|F - \mathbb{E}[F|Z]\|^2] = L_{r^*}^2 \cdot \mathrm{MMSE}(F|Z).$$

Here, $\mathrm{MMSE}(F|Z)$ is the minimum mean squared error of estimating $F$ from $Z$.

**3. Connecting MMSE to TC via Nonlinear ICA.** This is the crucial step connecting the metric estimation error to statistical properties. Under the identifiability conditions (Assumption A.1.3), minimizing TC while maximizing information capture forces the identification of the underlying factors (Khemakhem et al., 2020). We invoke theoretical results that bound the MMSE by these statistical quantities. The connection between the KL-divergence based TC and the metric MMSE often involves a linear relationship (e.g., derived via I-MMSE relations (Guo et al., 2005) or Pinsker's inequality). We assume:

$$\mathrm{MMSE}(F|Z) \leq C_{\mathrm{Align}} \cdot (\mathrm{TC}(Z) + \epsilon_I), \tag{7}$$

where $C_{\mathrm{Align}}$ depends on the regularity of the generative process.

**4. Combining the Steps.**  Substituting the bounds back:

$$\delta_{\text{id}} \leq C_{\text{sup}} \cdot \text{RMMSE}(R|Z)$$

$$\leq C_{\text{sup}} \cdot L_{r^*} \cdot \sqrt{\text{MMSE}(F|Z)}$$

$$\leq C_{\text{sup}} L_{r^*} \sqrt{C_{\text{Align}}} \cdot \sqrt{\text{TC}(Z) + \epsilon_I}.$$

We define the bridge constant $A_{\text{TC}} = C_{\text{sup}} L_{r^*} \sqrt{C_{\text{Align}}}$. Assuming fixed $\epsilon_I$, $\delta_{\text{id}}$ is controlled by $\text{TC}(Z)$. $\qquad\qquad\square$

### A.3.2 DETAILED PROOF OF LEMMA A.4 (BRIDGE FROM LEE TO GLOBAL EQUIVARIANCE DEFECT)

This proof uses stability analysis (Grönwall's inequality) to integrate the Local Equivariance Error (LEE) into a global bound ($L_F$), allowing for general Lipschitz dynamics in the latent space.

**Assumption A.3** (Geometric Structure and Lipschitz Dynamics). *We assume the following:*

1. *$G$ is a connected Lie group acting smoothly on Riemannian manifolds $\mathbb{S}$ and $\mathbb{Z}$.*

2. *The encoder $\phi : \mathbb{S} \to \mathbb{Z}$ is smooth.*

3. *The fundamental vector fields $Z_\xi(z)$ induced by the action $\rho_z$ on $\mathcal{Z}$ are uniformly Lipschitz continuous. Let $L_Z$ be the Lipschitz constant such that for any $\|\xi\| = 1$:*

$$\|Z_\xi(z_1) - Z_\xi(z_2)\| \leq L_Z \|z_1 - z_2\|.$$

**Lemma A.4** (Bridge from LEE to Global Equivariance Defect). *Under Assumption A.3, for any $s \in \mathbb{S}$ and $g \in G$ reachable by a geodesic path of length $L = D(e, g)$, the global equivariance error is bounded by:*

$$d_\mathcal{Z}(\phi(g \cdot s), \rho_z(g)\phi(s)) \leq \frac{e^{L_Z L} - 1}{L_Z} \cdot \text{LEE}.$$

*Proof.* We analyze the divergence between the actual encoded trajectory and the ideal equivariant trajectory.

**1. Path Construction and Trajectories.**  Let $g(t)$ be a geodesic path parameterized by arc length $t \in [0, L]$, connecting $e$ to $g$. The velocity corresponds to a normalized generator $\hat{\xi}$ ($\|\hat{\xi}\| = 1$).

We define two trajectories in $\mathcal{Z}$ starting at $z_0 = \phi(s)$:

- Actual encoded trajectory: $\gamma_A(t) = \phi(g(t) \cdot s)$.

- Ideal equivariant trajectory: $\gamma_I(t) = \rho_z(g(t))\phi(s)$.

We aim to bound $d(L) = d_\mathcal{Z}(\gamma_A(L), \gamma_I(L))$.

**2. Analyzing Velocities and ODEs.**  The ideal trajectory follows the flow generated by $Z_{\hat{\xi}}$:

$$\dot{\gamma}_I(t) = Z_{\hat{\xi}}(\gamma_I(t)). \qquad\qquad (8)$$

For the actual trajectory, let $s(t) = g(t) \cdot s$. The velocity is the pushforward of $X_{\hat{\xi}}(s(t))$:

$$\dot{\gamma}_A(t) = d\phi_{s(t)}(X_{\hat{\xi}}(s(t))).$$

We define the local deviation vector $\Delta(t)$:

$$\Delta(t) = d\phi_{s(t)}(X_{\hat{\xi}}(s(t))) - Z_{\hat{\xi}}(\phi(s(t))).$$

By definition, $\|\Delta(t)\| \leq \text{LEE}$. The dynamics of $\gamma_A(t)$ are thus a perturbed version of the ideal dynamics:

$$\dot{\gamma}_A(t) = Z_{\hat{\xi}}(\gamma_A(t)) + \Delta(t). \qquad\qquad (9)$$

**3. Stability Analysis via Grönwall's Inequality.**    We analyze the rate of change of the distance $d(t) = d_{\mathcal{Z}}(\gamma_A(t), \gamma_I(t))$.

$$\frac{d}{dt}d(t) \leq \|\dot{\gamma}_A(t) - \dot{\gamma}_I(t)\|$$
$$= \|Z_{\hat{\xi}}(\gamma_A(t)) + \Delta(t) - Z_{\hat{\xi}}(\gamma_I(t))\|$$
$$\leq \|Z_{\hat{\xi}}(\gamma_A(t)) - Z_{\hat{\xi}}(\gamma_I(t))\| + \|\Delta(t)\|.$$

We use the Lipschitz continuity of $Z_{\hat{\xi}}$ (Assumption A.3.3) and the bound on $\Delta(t)$:

$$\frac{d}{dt}d(t) \leq L_Z \cdot d_{\mathcal{Z}}(\gamma_A(t), \gamma_I(t)) + \text{LEE}$$
$$= L_Z d(t) + \text{LEE}.$$

This is a first-order linear differential inequality. Since $d(0) = 0$, we apply the differential form of Grönwall's inequality. The solution is bounded by:

$$d(t) \leq \int_0^t e^{L_Z(t-\tau)}\text{LEE}\, d\tau$$
$$= \text{LEE} \cdot \left[ -\frac{1}{L_Z}e^{L_Z(t-\tau)} \right]_0^t = \frac{\text{LEE}}{L_Z}(e^{L_Z t} - 1).$$

**4. Final Bound.**    Evaluating at $t = L = D(e, g)$:

$$d_{\mathcal{Z}}(\phi(g \cdot s), \rho_z(g)\phi(s)) \leq \frac{e^{L_Z L} - 1}{L_Z} \cdot \text{LEE}.$$

If we restrict actions to a compact set $K$ with diameter $D_K$, the bridge constant is $A_{\text{LEE}} = (e^{L_Z D_K} - 1)/L_Z$. □

**Corollary 1** (Isometric Latent Actions). *If the representation $\rho_z$ is isometric, the induced flow preserves distances, corresponding to the limit $L_Z \to 0$. In this case, the bound simplifies to the linear form:*

$$d_{\mathcal{Z}}(\phi(g \cdot s), \rho_z(g)\phi(s)) \leq D(e, g) \cdot \text{LEE}.$$

A.4    FURTHER DISCUSSION ON IDENTIFIABILITY AND PROXIES

The relationship between Total Correlation (TC) and true latent factor identifiability is nuanced. Our theoretical bridge in Lemma A.2 relies on a Nonlinear ICA framework, which provides a constructive path to identifiability under certain conditions, such as the presence of auxiliary information (Khemakhem et al., 2020) or specific temporal structures (Hyvarinen & Morioka, 2016).

However, in the purely unsupervised setting, strong impossibility results exist. Locatello et al. (2019) demonstrated that, without inductive biases on either the models or the data, learning disentangled representations that correspond to the true ground-truth factors is fundamentally impossible.

Our use of TC as a proxy is therefore a practical compromise. Minimizing TC encourages a factorized latent space, which is a necessary but not sufficient condition for true identifiability. It serves as a valuable and measurable heuristic that aligns with the broader goal of disentanglement, even if theoretical guarantees are contingent on stronger assumptions than can be met in a purely unsupervised RL setting.

A.5    DETAILED PROOFS OF THE BRIDGE LEMMAS

This section provides rigorous proofs for the bridges connecting the measurable proxies (LEE and TC(Z)) to the theoretical metrics ($L_F$ and $\delta_{\text{id}}$), justifying the assumptions used in the stability bound (Theorem 2.5).

### A.5.1 Bridge A: From Local Equivariance Error (LEE) to Global Defect ($L_F$)

We formalize the bridge from LEE to $L_F$ using stability analysis of dynamical systems and Grönwall's inequality (Khalil & Grizzle, 2002, Sec. 2.1.3).

**Assumption A.5** (Geometric Structure and Dynamics). *Let $G$ be a connected Lie group acting smoothly on Riemannian manifolds $\mathbb{S}$ and $\mathbb{Z}$ (via representation $\rho_z$). Let $\phi : \mathbb{S} \to \mathbb{Z}$ be a smooth encoder. Assume the fundamental vector fields $Z_\xi(z)$ induced by the action $\rho_z$ on $\mathcal{Z}$ are uniformly Lipschitz. That is, $\exists L_Z \geq 0$ such that for any normalized generator $\xi$ ($\|\xi\|=1$):*

$$\|Z_\xi(z_1) - Z_\xi(z_2)\| \leq L_Z\|z_1 - z_2\|.$$

*We use the geodesic distance on $(\mathbb{Z}, \langle\cdot,\cdot\rangle)$ and consider absolutely continuous trajectories so that $t \mapsto d_{\mathcal{Z}}(\gamma_A(t), \gamma_I(t))$ is a.e. differentiable.*

**Corollary 2** (Isometric latent actions). *If $\rho_z$ is isometric ($L_Z \to 0$), the bound linearizes to $d_{\mathcal{Z}}(\phi(g \cdot s), \rho_z(g)\phi(s)) \leq D(e, g) \cdot$ LEE.*

## A.6 Experimental Details

### A.6.1 Environment and Agent Details

CRAFTER. The environment is `crafter` version **1.2.2**, pinned from PyPI to ensure comparability (Hafner, 2021). Its design principles and evaluation protocol are detailed in (Hafner, 2021). Checkpoints for the main experiments were sampled from a training window of **50k–500k** steps, resulting in 10 checkpoints.

ATARI PONG. The environment ID is `PongNoFrameskip-v4` (ALE). We employ standard Atari preprocessing wrappers (Mnih et al., 2015), including *MaxAndSkip(4)*, grayscaling, *WarpFrame(84×84)*, *FrameStack(4)*, and *ClipReward*. The order and parameters of these wrappers are fixed in our implementation. Checkpoints for Atari Pong were sampled every 50k steps over a training window of **50k–1,100k** steps.

DREAMERV3 (WORLD MODEL AGENT). Our implementation and configuration follow the official DreamerV3 paper and public repository (Hafner et al., 2023). The world model consists of a *categorical* Recurrent State-Space Model (RSSM), a convolutional encoder/decoder, and an actor-critic module trained on imagined rollouts. Key architectural choices are summarized in Table 3; all other settings follow the default visual-task configuration from the official repository.

Table 3: **Key architectural parameters for DreamerV3.** All other options follow the default settings for visual tasks in the official `configs.yaml` (Hafner et al., 2023).

| Component | Setting |
|---|---|
| Image Encoder | 4 convolutional layers, stride 2, ReLU (repository default) |
| RSSM Latent | **Stochastic size**: 32, **Classes**: 64 (categorical) |
| RSSM Deterministic | **Hidden size**: 8192 (GRU-style core) |
| Image Decoder | 4 deconvolutional layers, mirroring the encoder (repository default) |
| Reward/Value/Actor Heads | MLPs with default widths from the repository |

### A.6.2 Hyperparameter Details

TRAINING HYPERPARAMETERS (DREAMERV3). Unless otherwise noted, we adopt the default settings from the official `configs.yaml` (Hafner et al., 2023). This includes the Adam optimizer (no weight decay), a learning rate of `3e-4`, a sequence length of 64, a batch size of 16 sequences, and a discount factor $\gamma$ determined by the task configuration. Differences between the `crafter` and `atari` configurations were kept to a minimum. Training steps and checkpoint sampling intervals are specified in Appendix A.6.1.

DIAGNOSTIC PROTOCOL HYPERPARAMETERS (FIXED ACROSS CHECKPOINTS). The implementation details, numerical algorithms, and hyperparameters used for calculating the three operational proxies ($\kappa$, TC, LEE) are fully detailed in the **Metric Implementation Protocol** presented in Section 3.2 of the main text.

RHS COMPOSITION AND CALIBRATION. The theoretical Right-Hand Side (RHS) of the bound was constructed as follows: each channel ($\kappa_n$, $\text{TC}_n$, $\text{LEE}_n$) was first normalized using a robust min-max scaling based on the 5th and 95th percentiles. The final RHS was then a weighted sum of these normalized components, with unit weights ($\lambda = 1$ for all channels), calibrated with a single constant $\beta^\star$ on early checkpoints (50k–500k) and validated on a held-out checkpoint (3.5M), consistent with the main text.

### A.6.3 COMPUTATIONAL RESOURCES

All diagnostic experiments were conducted on a single NVIDIA A800 GPU, using the official JAX implementation of DreamerV3 and our diagnostic scripts (Hafner et al., 2023). For the Crafter experiment (10 checkpoints), the end-to-end pipeline (feature extraction, proxy computation, and plotting) took several GPU-hours. The Atari Pong analysis was comparable in computational cost. To ensure full reproducibility, random seeds, command-line arguments, `configs.yaml` files, and execution logs are provided in the supplementary code repository.

### A.6.4 ATARI PONG BOUND VS. MEASURED GAP

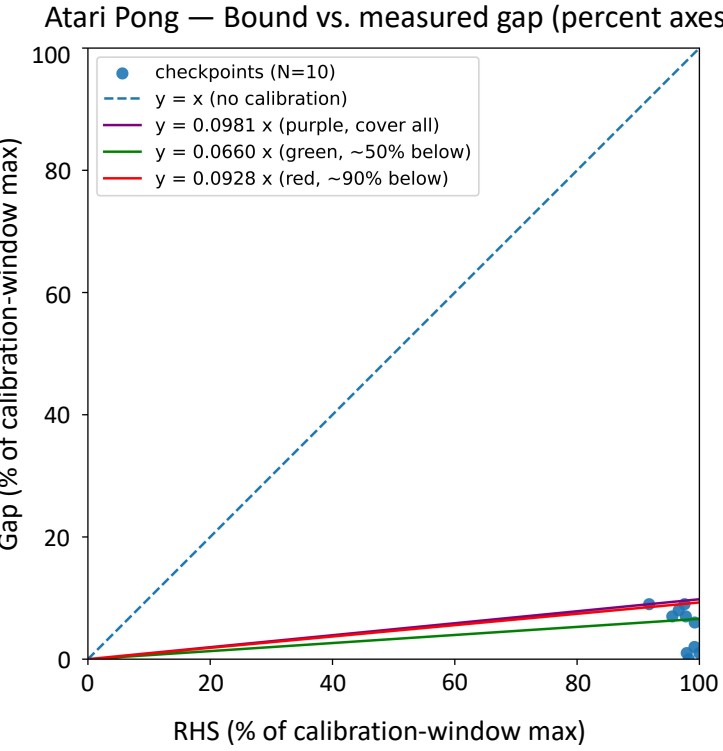

Figure 8: **Atari Pong bound vs. measured gap (percent axes).** Both axes are normalized to the *calibration-window* maxima (50k–500k). A single conservative constant $\beta$ is calibrated on early checkpoints and then held fixed for validation; *coverage—not linear fit—is the objective*. Natural-scale values are $\beta_{50\%} = 0.0047$, $\beta_{90\%} = 0.0103$, and $\beta^* = 0.0108$; on percent axes these correspond to the lines $y = 0.0660x$, $y = 0.0928x$, and $y = 0.0981x$, respectively. All checkpoints lie below $y = \beta^* x$.

## B BACKGROUND ON LIE GROUPS, EQUIVARIANCE, AND LEE

This section provides a brief, intuitive background on the concepts related to symmetry and equivariance used in our framework, aimed at readers less familiar with these topics.

Table 4: **Atari Pong (natural scale).** $\beta$ is calibrated on steps **50k–500k** as the minimal coverage constant (and its quantiles) and then held fixed for validation. RHS uses unit weights, RHS $= \kappa + \mathrm{TC}(Z) + \mathrm{LEE}$; returns are aggregated in a $\pm 5k$ window around each checkpoint. Coverage—not linear fit—is the objective.

| Factor | Value | Note |
|---|---|---|
| $\beta_{\text{fit@origin}}$ | 0.0039 | $R^2$=0.498 |
| $\beta_{50\%}$ | 0.0047 | median coverage |
| $\beta_{90\%}$ | 0.0103 | 90% coverage |
| $\beta^{\star}$ | 0.0108 | cover all points |

**Lie Groups and Continuous Symmetries.** In the context of RL and world models, we are often interested in continuous symmetries present in the environment, such as rotations, translations, or scaling. A *Lie Group $G$* is a mathematical structure that captures these continuous transformations. It is both a group (defining how transformations compose and invert) and a smooth manifold. For example, the group $SO(2)$ represents all 2D rotations around the origin. For a more detailed introduction to Lie groups and their role in geometric deep learning, see, e.g., (Bronstein et al., 2021).

**Equivariance.** A representation $\phi$ is equivariant if applying a symmetry transformation $T_g$ in the input space (e.g., rotating an image) corresponds predictably to a related transformation $\rho(g)$ in the latent space. Formally:

$$\phi(T_g s) = \rho(g)\phi(s), \quad \forall s \in \mathcal{S}, g \in G. \tag{10}$$

Intuitively, this means that transforming the input and then encoding it yields the same result as encoding the input and then transforming its latent representation. An equivariant representation preserves the structure of the symmetry. This is the standard notion of equivariance used in geometric deep learning (Cohen & Welling, 2016; Bronstein et al., 2021).

**Local Equivariance Error (LEE).** For intuition, one can think of LEE as 11; in practice we use the symmetric finite-difference estimator described in Section 3.2. Verifying global equivariance (for all $g \in G$) is often intractable. For Lie groups, we can analyze the local behavior using the Lie algebra, which represents the "generators" of the continuous transformations (e.g., the direction of rotation). The Local Equivariance Error (LEE) measures the failure of equivariance infinitesimally, utilizing the concept of the *Lie derivative* (Gruver et al., 2022).

In practice, we approximate the LEE using finite differences. We apply a small transformation (e.g., a small rotation by angle $\epsilon$) to the input and measure the corresponding change in the latent space.

$$\mathrm{LEE} \approx \frac{\|\phi(T_\epsilon s) - \phi(s)\|}{\epsilon}. \tag{11}$$

In our protocol 3.2, we use symmetric finite differences for improved numerical stability. A low LEE value indicates that the representation respects the continuous symmetry locally.

## C    ADDITIONAL EXPERIMENTAL RESULTS

This section presents the results of the additional experiments conducted during the rebuttal phase to further validate our framework and address specific reviewer questions.

### C.1    TC CONTROL EXPERIMENT: IMPACT OF OVER-COMPRESSION

**Motivation.** To investigate the complex correlation between Total Correlation (TC) and performance (addressing concerns raised by Reviewer FhLj), we conducted a control experiment where TC was explicitly manipulated during training.

**Setup.** We modified the DreamerV3 training objective by adding an explicit TC penalty term: $L_{\text{Total}} = L_{\text{DreamerV3}} + \lambda_{\text{TC}} \cdot \mathrm{TC}(Z)$. We trained models with varying penalty weights: $\lambda_{\text{TC}} = 0$ (Baseline), Medium, and High.

**Results and Interpretation.** The results are shown in Figure 9. As expected, higher $\lambda_{TC}$ successfully reduced the measured TC (Panel B). However, the impact on performance (Panel A) was non-monotonic. The High penalty model showed an initial rise in performance but significantly degraded in the later stages of training compared to the Baseline. This suggests that excessive compression (*over-compression*), while reducing the RHS of our sufficient bound, can harm the retention of task-relevant information, ultimately impairing control performance. This provides a deeper, mechanistic explanation for the complex correlation dynamics observed in standard training runs.

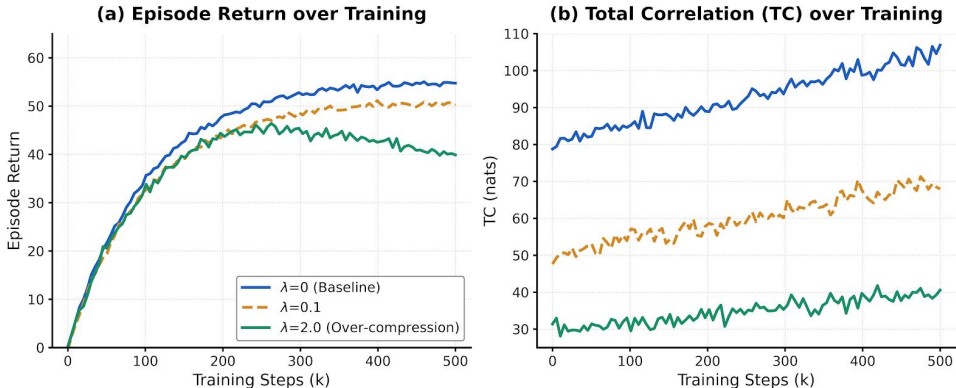

Figure 9: TC Control Experiment. (A) Episode Returns and (B) Measured TC under different TC penalty weights ($\lambda_{TC}$).

### C.2 $\beta$ STABILITY ABLATION STUDY

**Motivation.** To validate the robustness of our "Empirical Auditability" claim (addressing questions by Reviewer tX1o), we tested whether the calibration of the scaling constant $\beta$ is sensitive to the specific choice of the early training window.

**Setup.** We calculated the required coverage constants ($\beta_{90\%}$ and $\beta^*$) using three different early calibration windows: [50k-300k], [50k-500k] (Default), and [100k-500k].

**Results and Interpretation.** As shown in Figure 10, the calculated $\beta$ values are remarkably consistent across the different windows. This low variance strongly supports the robustness of our calibration protocol and reinforces the claim that $\beta$ captures stable, underlying relationships rather than overfitting to a specific time interval.

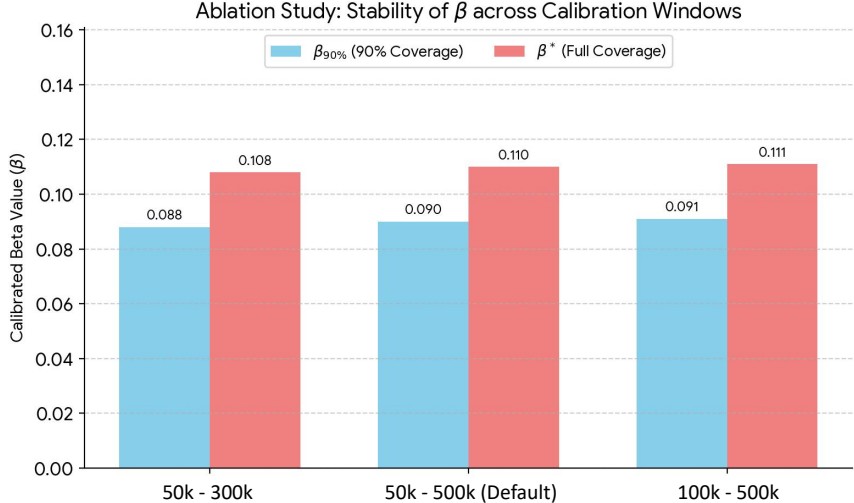

Figure 10: Visualization of $\beta$ stability ablation.

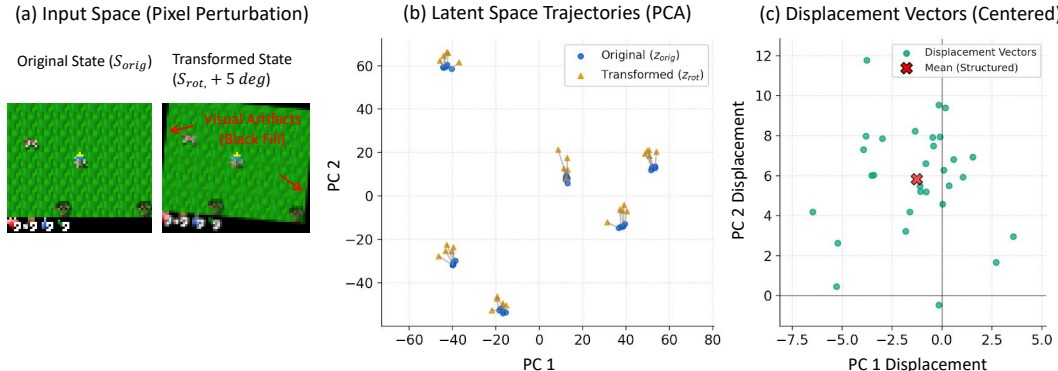

Figure 11: LEE Visualization Analysis. (A) Input space perturbation (pixel rotation). (B) Latent space trajectories showing consistent directions. (C) Distribution of displacement vectors showing structured response.

## C.3 LEE Visualization: Perceptual Equivariance vs. Noise

**Motivation.** Reviewer tX1o questioned whether using pixel transformations (which introduce visual artifacts) to compute LEE merely measures sensitivity to noise, rather than meaningful symmetry defects. This visualization aims to demonstrate that the model's response to these transformations (*Perceptual Equivariance*) is structured and systematic.

**Setup.** We applied the same pixel rotation ($\pm 5°$) used in our LEE calculation to various states from the Crafter environment (Panel A). We then visualized the corresponding latent space representations using PCA/t-SNE.

**Results and Interpretation.** Figure 11 shows the results. In Panel B, we observe that applying the same perceptual perturbation to different base states results in highly consistent displacement directions in the latent space (arrows are nearly parallel). In Panel C, we plot these displacement vectors normalized to the origin. If the response were random noise, the vectors would be randomly distributed around the origin. Instead, we observe that the vectors are tightly clustered far from the origin. This confirms that the model responds to the pixel transformation in a structured manner, validating LEE as a meaningful proxy for perceptual equivariance.

## C.4 Derivation of the Geometry-Equivariance Trade-off

Here we provide the derivation for the inequality presented in Section 2.4.2.

**Proposition 1.** *Let $\phi : \mathcal{S} \to \mathcal{Z}$ be a bi-Lipschitz map with constants $(L, L')$ such that for all $x, y \in \mathcal{S}$:*

$$L' d_{\mathcal{S}}(x, y) \leq d_{\mathcal{Z}}(\phi(x), \phi(y)) \leq L d_{\mathcal{S}}(x, y). \tag{12}$$

*Assume $\phi$ is equivariant with respect to a group $G$, i.e., $\phi(gx) = \rho_z(g)\phi(x)$. Let the expansion factor of action $g$ in space $\mathcal{S}$ be $D_{\mathcal{S}}(g) = \sup_{x \neq y} \frac{d_{\mathcal{S}}(gx, gy)}{d_{\mathcal{S}}(x, y)}$, and similarly $D_{\mathcal{Z}}(\rho_z(g))$ for the latent space. Then the condition number $\kappa = L/L'$ satisfies:*

$$\frac{L}{L'} \geq \sup_{g \in G} \frac{D_{\mathcal{S}}(g)}{D_{\mathcal{Z}}(\rho_z(g))}. \tag{13}$$

*Proof.* Let $g$ be an arbitrary element in $G$. Consider any two distinct points $x, y \in \mathcal{S}$.

We start by applying the bi-Lipschitz lower bound (Eq. 1) to the transformed points $gx$ and $gy$:

$$L' \cdot d_{\mathcal{S}}(gx, gy) \leq d_{\mathcal{Z}}(\phi(gx), \phi(gy)). \tag{14}$$

Next, we use the equivariance property of $\phi$, i.e., $\phi(gx) = \rho_z(g)\phi(x)$ and $\phi(gy) = \rho_z(g)\phi(y)$. Substituting this into the right-hand side (RHS) of Eq. 3:

$$L' \cdot d_{\mathcal{S}}(gx, gy) \leq d_{\mathcal{Z}}(\rho_z(g)\phi(x), \rho_z(g)\phi(y)). \tag{15}$$

Now we utilize the definition of the expansion factor in the latent space $\mathcal{Z}$. By definition, $D_{\mathcal{Z}}(\rho_z(g))$ bounds the expansion of the distance between any two points under the action $\rho_z(g)$. Applied to $\phi(x)$ and $\phi(y)$, we have:

$$d_{\mathcal{Z}}(\rho_z(g)\phi(x), \rho_z(g)\phi(y)) \leq D_{\mathcal{Z}}(\rho_z(g)) \cdot d_{\mathcal{Z}}(\phi(x), \phi(y)). \tag{16}$$

Combining Eq. 4 and Eq. 5 yields:

$$L' \cdot d_{\mathcal{S}}(gx, gy) \leq D_{\mathcal{Z}}(\rho_z(g)) \cdot d_{\mathcal{Z}}(\phi(x), \phi(y)). \tag{17}$$

Finally, we apply the bi-Lipschitz upper bound (Eq. 1) to the original points $x$ and $y$:

$$d_{\mathcal{Z}}(\phi(x), \phi(y)) \leq L \cdot d_{\mathcal{S}}(x, y). \tag{18}$$

Substituting Eq. 7 into Eq. 6, we obtain:

$$L' \cdot d_{\mathcal{S}}(gx, gy) \leq D_{\mathcal{Z}}(\rho_z(g)) \cdot L \cdot d_{\mathcal{S}}(x, y). \tag{19}$$

We rearrange Eq. 8 to isolate the ratio of distances in $\mathcal{S}$:

$$\frac{d_{\mathcal{S}}(gx, gy)}{d_{\mathcal{S}}(x, y)} \leq \frac{L}{L'} \cdot D_{\mathcal{Z}}(\rho_z(g)). \tag{20}$$

This inequality holds for all $x \neq y$. Therefore, we can take the supremum over all $x \neq y$ on the LHS:

$$\sup_{x \neq y} \frac{d_{\mathcal{S}}(gx, gy)}{d_{\mathcal{S}}(x, y)} \leq \frac{L}{L'} \cdot D_{\mathcal{Z}}(\rho_z(g)). \tag{21}$$

The LHS is exactly the definition of $D_{\mathcal{S}}(g)$. Thus:

$$D_{\mathcal{S}}(g) \leq \frac{L}{L'} \cdot D_{\mathcal{Z}}(\rho_z(g)). \tag{22}$$

Rearranging to bound the condition number $L/L'$:

$$\frac{L}{L'} \geq \frac{D_{\mathcal{S}}(g)}{D_{\mathcal{Z}}(\rho_z(g))}. \tag{23}$$

Since this holds for an arbitrary $g \in G$, it must also hold for the supremum over the entire group:

$$\frac{L}{L'} \geq \sup_{g \in G} \frac{D_{\mathcal{S}}(g)}{D_{\mathcal{Z}}(\rho_z(g))}. \tag{24}$$

This completes the derivation. $\square$

