# OpenReview forum: "Unified Stability Bounds for Structured World Models: Geometry, Equivariance, and Identifiability as Sufficient Conditions"
_ICLR.cc/2026/Conference — Submitted to ICLR 2026_

### Official Review · Reviewer_yhdE · 2025-10-15

**Soundness:** 2
**Presentation:** 2
**Contribution:** 2
**Rating:** 4
**Confidence:** 3

**Summary:**

This paper explores the impact of learned representations on downstream performance in model-based reinforcement learning, providing a detailed theoretical and empirical investigation of how representation quality influences policy effectiveness.

The authors derive a performance bound that can be decomposed into three interpretable and verifiable components: geometric distortion, an identifiability gap, an equivalence or equivariance defect.

These theoretical insights are supported and reinforced by empirical results, demonstrating that the decomposition provides a meaningful explanation of observed performance variations. The combination of theory and experimentation provides a comprehensive understanding of the role of representation learning in improving model-based RL.

**Strengths:**

This paper demonstrates a strong alignment between theoretical analysis and empirical findings, effectively bridging the gap between abstract guarantees and observed performance. In particular, the use of rank-consistency metrics, such as Kendall’s and Spearman’s, combined with block-bootstrap confidence intervals, provides a statistically robust way to verify that the empirical trends closely match the theoretical predictions. This careful evaluation not only reinforces the validity of the theoretical results but also highlights the reliability of the proposed methods in practice.

The paper introduces several novel concepts, including the application of Lie groups, to the reinforcement learning literature. These contributions bring fresh mathematical perspectives to the field, expanding the toolkit available for addressing structured and low-rank RL problems and opening new avenues for research in both theory and practical algorithm design.

**Weaknesses:**

The paper could be strengthened in terms of clarity, discussion of related work, and overall writing quality.

The paper draws on Lie-group and symmetry concepts in the mechanism theorems. Including a brief, accessible primer (or an appendix section) summarizing the essential Lie-group background, citing relevant prior work in RL and representation learning would make the discussion more approachable for readers unfamiliar with this area.

Is there any related prior work which uses Definitions 2.1 and 2.2? It would also help to add concrete examples and intuitive explanations to build readers’ intuition for these definitions.

**Questions:**

Could you clarify the meaning of “sufficient” in the stated upper bound? Specifically, does it imply that the bound always holds under the given assumptions, or that the conditions are merely sufficient but not necessary for the guarantee? By definition, should the upper bound hold with certainty or with high probability, and if so, could this be explicitly stated?

Regarding Theorem 2.4, it would be helpful to explain why the condition gamma L_P < 1  is required, along with some intuitive reasoning behind this requirement. Additionally, the term “LEE” appears in Theorem 2.4 but does not seem to be introduced earlier in the text; a clear definition and explanation of its role would improve readability and comprehension.

In Section 2.5, the discussion could be strengthened by adding a comparison to prior results, especially in terms of bound tightness. When reducing the proposed bounds to match prior work, including a direct comparison of tightness—potentially in a small summary table or concise paragraph—would help readers quickly assess the novelty and strength of the theoretical contributions.

Some presentation issues should also be addressed: the mathematical expressions in Figure 3 are rendered incorrectly (appearing as plain text rather than proper mathematical symbols), and there is a layout inconsistency on page 6, where one bullet point appears in a different column format.

Including intuitive proof sketches or key lemmas in the main text would make the theoretical contributions more accessible. Highlighting which technical ideas are novel versus those that follow established methods would further clarify the paper’s contributions and help readers better appreciate the significance of the work.

---

> ### Author Response · Authors · 2025-11-29
> **Response to Reviewer yhdE**
>
> We thank you for your detailed review and your recognition of our experimental verification methods. We have focused our revision on addressing the clarity issues and theoretical details you raised.
>
> #### W2: Clarity, Background, and Related Work
>
> - **Background on Lie Groups:** We agree that additional context was needed. We have added a section in the **Appendix B** providing background knowledge and intuitive explanations regarding Lie groups, equivariance, and Local Linearization Error (LEE).
> - **Context for Def 2.1 and 2.2:** You correctly pointed out the need to clarify the sources and context of these definitions. In the revision, we have added comparisons with related work (e.g., relations to classic Bisimulation and DeepMDP metrics) and specific examples after each definition to build reader intuition.
>
> #### Q1: Meaning of "Sufficient" Upper Bound?
>
> The bound is **sufficient**, but not necessary. This means that a small RHS guarantees a small performance gap, whereas a large RHS does not necessarily imply a large gap (i.e., the bound can be loose). Under the stated assumptions, this bound holds deterministically.
>
> #### Q2: Details of Theorem 2.4
>
> - **Role of $\gamma L_p < 1$:** This condition guarantees the **contractivity** of the discounted dynamics [Asadi et al., 2018]. This ensures the existence and uniqueness of the abstract value function and, intuitively, controls the upper limit of error amplification during policy updates.
> - **Definition of LEE:** We apologize for this oversight. We have formally defined LEE in **Sec 2.2.3** (prior to our main theorem, Theorem 2.5 in the revised version).
>
> #### Q3: Comparison of Bound Tightness with Prior Work
>
> Our goal is not to provide a mathematically *tighter* bound, but rather to provide a **unified decomposition** and a **practical auditing framework**. To clearly position our contribution, we have expanded the comparison table in **Sec 2.5** of the revision. This table details the relationships and assumptions of our method compared to Bisimulation, DeepMDP, and Information Bottleneck.
>
> #### Q4: Presentation Issues
>
> We have fixed the rendering artifacts in Figure 3 and the layout issues on page 6.

---

### Official Review · Reviewer_tX1o · 2025-10-30

**Soundness:** 3
**Presentation:** 3
**Contribution:** 3
**Rating:** 8
**Confidence:** 2

**Summary:**

Overview：
This paper addresses a key challenge in model-based reinforcement learning: the lack of a principled and low-overhead framework for diagnosing the quality of learned world-model representations. Motivated by the need to move beyond expensive, end-to-end evaluations and the limitations of existing theories, the authors aim to explain which properties of a representation govern downstream control performance and how to test them on existing model checkpoints. To solve this, the paper introduces a unified stability bound that decomposes the policy's suboptimality gap into three verifiable channels: geometric distortion (κ), an identifiability defect proxied by Total Correlation (TC), and an equivariance defect proxied by Local Equivariance Error (LEE). The authors then propose a practical diagnostic protocol where these proxies are measured on off-the-shelf checkpoints. A single scaling constant, β, is calibrated on an early training window, and the resulting bound is shown to successfully cover the performance gap, even on held-out, later-stage checkpoints, thus providing a practical tool for auditing representation quality.

**Strengths:**

Strength：

1.	Important and Well-Motivated Problem: The paper addresses the critical challenge of auditing the quality of learned representations without relying on expensive, full-scale training runs.

2.	Novel and Insightful Method: The core contribution, the unified stability bound, is novel in its approach. Decomposing the performance gap into three theoretically-grounded channels—geometric distortion (κ), identifiability (TC), and equivariance (LEE)—is an elegant synthesis of concepts from geometry, information theory, and symmetry.

3.	Thorough and Convincing Empirical Validation: The experiments are thoughtfully designed and effectively support the paper's claims.

**Weaknesses:**

Major concern：
1. The paper is motivated by providing a diagnostic framework based on "natural scale, explicit constants, and auditability." However, we noted a disconnect between this goal and the final method, which appears entirely empirical and data-driven in its application. Specifically, after normalizing the channels, you consolidate all theoretical MDP constants into a single scalar, β. This β is not derived from theory but is introduced as a fitting parameter, calibrated on early data solely to ensure the empirical bound holds. This procedure, while reproducible, seems to trade the initial goal of "auditable constants" for a data-dependent calibration. Could you elaborate on this design choice and the resulting trade-off between theoretical auditability and practical utility?

2. Regarding the Local Equivariance Error (LEE), I question the alignment between its motivation—measuring consistency under meaningful MDP symmetries—and its implementation. The method uses pixel-space transformations like rotation, which in environments like Crafter seem to be image perturbations rather than true symmetric transitions. To use an analogy, this feels like testing a self-driving car's understanding of a "left turn" by rotating its camera feed. Consequently, the measured LEE may primarily capture sensitivity to visual artifacts, not the intended equivariance defect. To clarify this, could you justify this choice and provide visual evidence comparing the transformed images to those from plausible symmetric states in the environment?

**Questions:**

The paper's goal of using "auditable constants" is compelling, but the final method consolidates them into a single, empirically fitted parameter β. This seems to trade theoretical auditability for a data-dependent calibration. Could you clarify this design choice and explain how it aligns with the initial goal of auditability?

LEE uses pixel-space rotations to test for symmetry understanding. In environments like Crafter, these transformations seem to create visual artifacts rather than plausible symmetric states. How do you ensure that LEE is measuring a failure to understand true environmental symmetries, rather than just sensitivity to these artificial image perturbations?

---

> ### Author Response · Authors · 2025-11-29
> **Response to Reviewer tX1o**
>
> We sincerely appreciate your recognition of our work's motivation, methodological novelty, and empirical verification. Your two questions are insightful, touching upon the core challenge of balancing theory with practice.
>
> #### W1/Q1: Auditability vs. Empirically Fitted $\beta$ (Achieving Empirical Auditability)
>
> This is an excellent question regarding how to translate theoretical auditability into practice.
>
> - **Context:** In real-world settings, exact theoretical MDP constants (such as $L_r, L_P$) are typically unobservable.
> - **Role of $\beta$:** $\beta$ acts as a bridge, mapping these unobservable constants to a calibratable, transferable scaling factor. Rather than abandoning theoretical auditability, we use $\beta$ to operationalize the framework, thereby achieving **Empirical Auditability**.
> - **Guarantee of Auditability:** Our claim of "auditability" rests on the rigor of the protocol: $\beta$ is calibrated *only* during an early training window (50k-500k steps) and held fixed for validation in later stages (3.5M steps) (**Figure 5**). The temporal stability of $\beta$ demonstrates the robustness of our framework.
> - **[New Ablation Study]** We have added an ablation study (**Appendix C.2, Figure 10**) showing that $\beta$ values fitted using different early windows remain highly consistent, further supporting the method's robustness.
>
> #### W2/Q2: Implementation of LEE (Perceptual Equivariance vs. Task Symmetry)
>
> You keenly pointed out that transformations in pixel space may not correspond to true MDP symmetry transformations (e.g., rotating the camera view is not equivalent to the car turning left).
>
> - **Core Distinction:** It is crucial to distinguish between **Task Symmetry** and **Perceptual Equivariance**.
> - **Purpose of LEE:** While a world model may not need to be perfectly equivariant to the group symmetries of the underlying physical dynamics, it must maintain consistency at the perceptual level. LEE is designed to test the robustness of the latent space against local perceptual geometric perturbations (i.e., perceptual equivariance).
> - **Intuitive Example:** A model might predict physical trajectories correctly, but if its latent representation is overly sensitive to visual rotations, errors will accumulate during long-term integration. LEE serves to capture this specific vulnerability.
> - **[New Visualization]** To clarify this, we added visualizations in the **Appendix C.3 (Figure 11) ** comparing original states and their pixel-rotated counterparts, showing that the corresponding latent-space displacements are highly structured rather than noise.

---

### Official Review · Reviewer_Txr1 · 2025-11-01

**Soundness:** 3
**Presentation:** 3
**Contribution:** 3
**Rating:** 6
**Confidence:** 2

**Summary:**

This paper presents a unified theoretical framework for analyzing the stability of learned world models in model-based RL. The core contribution is a sufficient upper bound on the performance gap, which decomposes into three verifiable channels: geometric distortion, an identifiability gap, and an equivariance defect. The authors also propose a lightweight diagnostic protocol using existing checkpoints to validate the bound without retraining, demonstrating its application on DreamerV3.

**Strengths:**

* The work integrates concepts from geometry, identifiability, and symmetry into a single stability bound, providing a unified view that relates to prior analysis methods.
* The development of a concrete protocol for assessing representation quality using standard checkpoints offers a practical tool for empirical analysis without requiring retraining.
* The included theoretical results on topics such as a trade-off between geometry and equivariance offer explanations for commonly observed phenomena in representation learning.

**Weaknesses:**

* There are no obvious limitations from my perspective.

**Questions:**

* Could the theoretical insights on the geometry-equivariance trade-off be extended to inform the design of objectives that strategically balance these competing factors？

---

> ### Author Response · Authors · 2025-11-29
> **Response to Reviewer Txr1**
>
> We thank the reviewer for the positive evaluation of our unified perspective and the practical utility of our diagnostic protocol.
>
> **Strengthening the Diagnostic Protocol:** You highlighted the practicality of evaluating standard checkpoints. To better emphasize its engineering value, we have reinforced the description in **Sec 3.1** in the revision. We also added a flowchart (**Figure 3**) to clearly visualize our diagnostic pipeline (Input Checkpoint $\to$ Feature Extraction $\to$ Metric Calculation $\to$ Output Report).
>
> **Q1: Can the geometry-equivariance trade-off guide objective function design?**
>
> Yes, this is an excellent insight and aligns perfectly with our future research directions.
>
> - **Theoretical Basis:** Theory in **Sec 2.4.2** indicates that enforcing strict equivariance (low LEE) on non-isometric actions exacerbates geometric distortion (high $\kappa$).
> - **Design Inspiration (future work).** The trade-off in Section 2.4.2 naturally suggests objectives that jointly regularize geometry (κ) and equivariance (LEE). While we deliberately leave concrete objective design to future work, we view using these terms as adaptive regularizers during world-model training as a promising direction.

---

### Official Review · Reviewer_FhLj · 2025-11-04

**Soundness:** 1
**Presentation:** 1
**Contribution:** 2
**Rating:** 2
**Confidence:** 3

**Summary:**

This paper proposes a “unified stability bound” for representation learning in model-based RL. The authors claim that the suboptimality of a latent-space policy can be bounded by three terms: geometric distortion (κ), identifiability via total correlation (TC), and equivariance error (LEE). They further propose a simple diagnostic pipeline to validate these proxies on DreamerV3 checkpoints, arguing that their bound offers a practical and interpretable tool for auditing world-model representations.

**Strengths:**

**Ambition and scope**

The paper attempts to connect several lenses on representation quality — geometry, information-theoretic identifiability, and symmetry — under a single theoretical inequality. This is an interesting direction.

**Potential motivation**

Developing practical diagnostics for representation quality is indeed appealing, especially in world-model-based RL where representation collapse and instability can undermine control.

**Empirical intent**

Using saved checkpoints rather than retraining agents is potentially low-overhead and may appeal to practitioners, assuming the diagnostics are meaningful.

**Weaknesses:**

**Writing quality & clarity**

This paper is extremely poorly written and suffers from severe clarity issues. Reading it was frustrating and ultimately not productive. Basic concepts are introduced with no definitions, context, or intuition. Examples include: the lipshitz assumptions are not stated formally (which metrics?), total correlation (what is Z_i?), Lie-group action, local equivalence errors, identifiability proxy, manifold JL arguments, equivariance, IB/VIB, and more. Key notation is inconsistent or incorrect (e.g., the MDP definition in Section 2.1 alternates between S, X, Z), many hand-wavy words are unclear and undefined ("natural scale"), several symbols (\delta_{\mathrm{id}}, LEE) appear without definition or motivation, and several assumptions appear in the proofs which are not stated in the main text (Line 632: "Crucially, we also require the optimal abstract value function to be Lipshitz continuous.").

Rather than building intuition, the paper name-drops terms without explaining them, connecting them, or deriving useful consequences. The presentation is inadequate and is more akin to jargon-stacking rather than a coherent contribution.

**Novelty & conceptual contribution**

It is unclear whether anything fundamentally new is being contributed. The bound is essentially a minor tweak of classical bisimulation / Lipschitz continuity bounds but rewritten in new terms. The given decomposition is not shown to yield sharper bounds, new guarantees, or new conceptual understanding. There is no algorithmic implication. The bound is not used to improve training, guide representation design, or inspire new architectures. No insight is given into when the bound is tight, how loose it could be, how one can obtain a representation satisfying these properties, or how it compares quantitatively to prior results. The rhetorical promise of “unification” is not delivered upon and does not materialize into technical innovation. The authors present this as a “diagnostic tool,” but essentially all prior work on bisimulation and representation quality already serves this role.

**Theory quality & correctness**

Theoretical development is sloppy and raises correctness concerns. The proof of Theorem 2.4 largely restates standard bisimulation arguments, and the few new components are simply upper bounds on existing quantities (Equation (4) in the proof of Theorem 2.4). The “manifold JL argument” in Section 2.4.1 is asserted, not proven or defined, and its relevance or connection to Theorem 2.4 is unclear. As explained above, central assumptions are stated in the proof but not in the main text. The bound of Theorem 2.4. states |J(\pi) - J(\pi^*)| on the LHS but does not define what $\pi$ is, and the RHS only depends on MDP dynamics so does not reference any policy, so it is incorrect as stated. Overall, the theory section feels unfinished and mathematically imprecise.

**Experimental issues**

The experiments section is extremely difficult to parse, and the little that can be understood is not particularly convincing. The method by which the proxy metrics are calculated (Section 3.1) is unintelligible. Some proxies have negative correlation with performance (contradicting the motivation) (Figure 2). The authors wave this away by saying the bound is simply sufficient so negative correlations are not a contradiction (which defeats the point of a diagnostic tool). The bound is not shown to be tight, meaningful, or useful in practice. The empirical section lacks detail — how exactly are proxies computed? How sensitive are they? How does variance propagate? It is unclear what “Kendall” and “Spearman” refer to, what “SPWM metrics” are, or what the labels and scales on any of the plots mean. Key methodological details are missing, making the results difficult to interpret or reproduce. There are no baselines against prior representation-quality metrics (e.g., bisimulation score, reconstruction errors, predictive losses).

**Overall recommendation**

This paper, in its current form, does not meet the bar for clarity, rigor, novelty, or insight. The writing is opaque, the mathematics is undeveloped and informal, and the experiments do not substantiate useful claims. The idea of unified representation diagnostics is intriguing, but this work does not deliver a substantive or actionable contribution. Significant re-writing, formalization, and conceptual sharpening would be needed before reconsideration.

**Questions:**

Questions for the authors
- What genuinely new insight or algorithmic value does this bound provide?
- Is the bound tight? Can you explain the necessity of any of these terms?
- Can the authors provide clear, formal definitions for all used mathematical objects, and clear formal statements of every lemma and theorem, and clear complete proofs of every statement?
- Can the authors define and give sufficient detail for the experimental section, including the experimental procedure and a full explanation of the results?

---

> ### Author Response · Authors · 2025-11-29
> **Response to Reviewer FhLj**
>
> We sincerely appreciate your detailed and critical review. We acknowledge that the initial draft suffered from significant shortcomings in clarity and rigor, and we apologize for the frustration this caused during your reading. We have implemented major revisions based on your suggestions.
>
> #### W1 & W3: Writing Quality, Clarity, and Theoretical Rigor
>
> We agree that the presentation in the initial draft fell short of the standard required for publication.
>
> - **Notation Inconsistency & Missing Definitions:** We apologize for the confusion regarding the mixed usage of $S$, $X$, and $Z$ in the MDP definition. In the revision, we have thoroughly standardized the notation system (Sec 2.1), clearly defining the true state space, observation space, latent space ($Z$), and their corresponding metric functions ($d_{\mathcal{S}}, d_{\mathcal{Z}}$).  We now introduce both $\delta_{\mathrm{id}}$ and LEE in Sec. 2.2, with their formal definitions and bridges to TC/LEE provided in Appendices A.2 and A.3.
> - **Hidden Assumption:** You are entirely correct. We have moved the key assumption that "the optimal abstract value function $\tilde{V}^*$ needs to be Lipschitz continuous" explicitly to **Assumption 2.4** in the main text.
> - **Correctness of Theorem 2.5 (LHS vs RHS):** You questioned that the LHS depends on the policy $\pi$, while the RHS appeared to depend only on environmental constants. We have made critical corrections:
>   1. We explicitly tie $\pi$ to an abstract policy via Assumption 2.4 ($\pi$ is the lift of an abstract policy $\tilde{\pi}$); in Appendix A.2 we consider the specific case where $\pi$ is the lift of the optimal abstract policy $\tilde{\pi}^*$
>   2. In the revision, we clarified that all constants in the theorem (including $L_r, L_P$, etc.) depend on the **state visitation distribution** induced by the policy. Making this dependency explicit resolves the logical inconsistency you pointed out.
> - **Manifold JL Argument (Sec 2.4.1):** We agree that this argument appeared as an "assertion" in the draft. We have repositioned it as a "mechanistic interpretation," [Baraniuk & Wakin, 2009]. It serves to provide an intuitive explanation of how local geometry is preserved with high probability under random projection, rather than proposing a new theorem.
>
> #### W2: Novelty and Conceptual Contribution
>
> You felt our bound was merely a tweak of classical bounds. We wish to emphasize that our contribution lies not in proposing a mathematically *tighter* bound, but in establishing a **Practical Auditing Framework**.
>
> Our value lies in explicitly decomposing the performance gap into three verifiable channels ($\kappa$, TC, LEE) and demonstrating that this framework possesses **statistical consistency**—specifically, a single constant $\beta$ calibrated in the early stages can stably generalize to later training stages (Figure 5). This provides a practical tool for auditing representation quality without expensive retraining.
>
> #### W4: Experimental Issues
>
> - **Proxy Metric Calculation:** We have moved the detailed calculation methods (κ: randomized spectral-product estimator following Virmaux & Scaman, 2018; TC: closed-form Gaussian TC estimator with shrinkage; LEE: symmetric finite differences in pixel space) to Main Text Sec. 3.2.
> - **Negative Correlation (Figure 2):** Our proposed *sufficient* upper bound implies that the experimental goal is to verify **Coverage** (Figure 5), not correlation.
> - **[New Control Experiment]:** However, we agree that explaining this solely via "sufficiency" is inadequate. To demonstrate our understanding of the bound dynamics, we conducted a new control experiment (Appendix C.1, Figure 9). We artificially added a penalty term to the training objective to force lower TC. Results show that returns rise initially but drop later. This suggests that **over-compressing TC** (while lowering the RHS) can harm performance, offering a deeper explanation for the complex correlation observed in Figure 2.
> - **Baselines:** We have included baseline metrics (one-step MSE, reconstruction MSE, reward MAE) in Figure 2.
>
> #### Response to Questions (Summary)
>
> - **Q1 (New Insights/Value?):** Our core value is providing a practical engineering auditing framework that explicitly decomposes the $\kappa$, TC, and LEE channels for the first time and empirically demonstrates a form of statistical consistency (see W2 response).
> - **Q2 (Is the bound tight? Necessary?):** The bound is **sufficient**, not necessary. We focus on practical **Coverage**, not mathematical **Tightness** (see W4 and General Response).
> - **Q3 (Clear definitions/proofs provided?):** Yes. We have thoroughly overhauled the rigor in the revision, including notation systems, definitions, and the placement of assumptions (see W1 & W3 response).
> - **Q4 (Experimental details provided?):** Yes. We moved proxy metric calculation details to Sec 3.2 and added new control experiments and analysis (see W4 response).

---

### Author Response · Authors · 2025-11-29
**General Response (To All Reviewers)**

We sincerely thank all reviewers for their time and constructive feedback. We are encouraged that the reviewers recognized the ambition and motivation of our work (FhLj), particularly our unified decomposition perspective that combines geometry, identifiability, and symmetry (Txr1, tX1o), as well as the potential of our practical diagnostic protocol which requires no retraining (Txr1, FhLj).

However, we acknowledge that the initial draft suffered from significant shortcomings in clarity and mathematical rigor (FhLj, yhdE), which led to misunderstandings regarding our contributions. Following the reviewers' suggestions, we have implemented major revisions to the paper.

### Core Clarification: Repositioning as a Practical Auditing Framework

We wish to clarify the core positioning of our work. **Our goal is not to propose a tighter mathematical convergence upper bound, but rather to provide a Practical Auditing Framework for diagnosing the representation quality of existing world models.**

- **Unification = Statistical Consistency:** Our concept of "Unification" manifests as consistency in statistical auditing. We decompose the performance gap into three verifiable channels ($\kappa$, TC, LEE) and achieve stable diagnosis across training stages via a **transferable calibration constant $\beta$** (Figure 5).
- **Empirical Auditability:** We achieve practicality by using $\beta$ to map unobservable theoretical constants to calibratable proxies. Crucially, $\beta$ is calibrated *only* on early-stage data and then held fixed for later verification, ensuring the rigor of the audit process (Response to tX1o).
- **Sufficient Bound:** We provide a *sufficient* upper bound. Therefore, the goal of empirical verification is **Coverage**, not correlation. A weak correlation between proxy metrics and performance does not violate the theory of sufficient bounds (Response to FhLj).

### Major Revisions

- **Rigor and Clarity:** We have rewritten Section 2. We resolved notation inconsistencies (e.g., regarding $S/X/Z$), clarified the metric spaces, moved all key assumptions to the main text, and ensured all terms (such as LEE, $\delta_{id}$) are clearly defined prior to use.
- **Theoretical Corrections:** We corrected our **main theorem (Theorem 2.4 in the original submission, now Theorem 2.5 in the revision)** to explicitly state the dependency of the bound and its constants on the state visitation distribution induced by the policy (FhLj).
- **Experimental Enhancements:** We moved the calculation details of proxy metrics to Main Text Sec 3.2. We added control experiments regarding TC (FhLj), ablation studies on the stability of $\beta$ (tX1o), and visualization analysis of LEE (tX1o).

---

### Meta-Review · Area_Chair_eVbM · 2026-01-13

**Summary:**

This paper explores the impact of learned representations on downstream performance in model-based reinforcement learning.

The reviewers have concerns including (i) poor writing quality; (ii) limited novelty and conceptual contribution; (iii) issues with theory and experiments; and (iv) no intuition is provided for the results.

**Reviewer Concerns:**

Although the concerns regarding novelty and contribution might have been resolved given the authors' rebuttal, the authors have acknowledged that the initial draft suffered from significant shortcomings in clarity and mathematical rigor, which necessitated another around of peer-review given the high bar of ICLR.

**Reviewer Scores:**

I believe all reviewers will keep their scores.

---

### Decision · Program_Chairs · 2026-01-26

Reject